

# Emerging mobile lidar technology to study boundary-layer winds influenced by operating turbines

Yelena Pichugina[1, 2], Alan W. Brewer[2], Sunil Baidar[1, 2] Robert Banta[1, 2], Edward Strobach[3], Brandi McCarty[1 ,2], Brian Carroll[1, 2], Nicola Bodini[4], Stefano Letizia[4], Richard Marchbanks[1, 2], Michael Zucker[1, 2], Maxwell Holloway[1, 2], and Patrick Moriarty[4]

[1]*CIRES, University of Colorado Boulder, Boulder, CO, USA*
[2]*NOAA Chemical Sciences Laboratory, Boulder, CO, USA*
[3]*AOSC University of Maryland, College Park, MD, USA*
[4]*National Renewable Energy Laboratory, Golden, CO, USA*

*Correspondence to*: Yelena L. Pichugina (Yelena.Pichugina@colorado.edu)

**Abstract.** The development of a microjoule-class pulsed Doppler lidar and deployment of this compact system on mobile platforms such as aircraft, ships, or trucks has opened a new opportunity to characterize the dynamics of complex mesoscale wind flows. The PickUp-based Mobile Atmospheric Sounder (PUMAS) truck-based lidar system was recently used during the American Wake Experiment (AWAKEN) to assess the general structure of boundary-layer wind and turbulence around wind turbines in central Oklahoma.

Wind speed profiles averaged over PUMAS transects influenced by the operating turbines (waked flow) show a 1–2 m s$^{-1}$ reduction compared to mean undisturbed (free flow) wind speed profiles. Spatial variability of wind speed was observed in time-height cross sections at different distances from turbines. The wind speeds were about 9–12 m s$^{-1}$ at 6 km distance compared to 5–7 m s$^{-1}$ at the transects near the turbines.

The PUMAS dataset from AWAKEN demonstrated the capability of the mobile Doppler lidar system to document spatial variability of wind flows at different distances from wind turbines and obtain quantitative estimates of wind speed reduction in the waked flow. The high-frequency, simultaneous measurements of the horizontal and vertical winds provide a new approach for characterizing dynamic processes critical for wind farm wake analyses.



## 1. Introduction

Stationary scanning Doppler lidars are a powerful remote sensing instrument that provide
high-quality measurements of wind and turbulence profiles from the surface up to several hundred
meters in the boundary layer. The Atmospheric Remote Sensing (ARS) group at the Chemical
Sciences Laboratory (CSL) of the National Oceanic and Atmospheric Administration (NOAA) uses
both commercial Doppler lidars and lidars developed within the group (Brewer and Hardesty,
1995). Lidar development at CSL goes back decades (Post and Cupp 1990_, Grund et al. 2002),
with continuous engineering updates and the design of new versions to meet research objectives.
Research studies on land using stationary scanning Doppler lidar have demonstrated the ability of
this instrument to reveal the structure and evolution of meteorological processes at a high vertical,
horizontal, and temporal resolution. Doppler lidar data are used to provide insight into boundary-
layer behavior during nocturnal stable and low-level jet (LLJ) conditions, among the most difficult
to characterize, understand, and model (Banta et al. 2003, 2006, Pichugina et al. 2010, 2023; Sun
et al. 2012). The lidar's three-dimensional (3D) scanning capability has been used to characterize
wind turbine wake properties and their downwind evolution, which is an important task for
optimizing wind farm layouts and power output. (Aitken et al. 2014; Banta et al. 2015, Bingöl et
al., 2010, Smalikho et al. 2013).
During the second Wind Forecast Improvement Project experiment, three scanning Doppler
lidars were deployed to the Columbia River Gorge to support the evaluation of the High Resolution
Rapid Refresh (HRRR) model, improve the prediction of winds in complex terrain (Olson et al.
2019; Banta et al. 2023, Pichugina et al, 2020, 2022), and to study wakes from the wind farm located
in the area (Wilczak et al. 2019). These studies used data from stationary Doppler lidars.
Motion-compensated lidar measurements from a mobile platform were obtained from a
NOAA research vessel in the Gulf of Maine. During these marine operations, the lidar was deployed
in a large seatainer with a GPS-based inertial navigation unit capable of determining platform
motion and orientation (Pichugina et al. 2012). A hemispheric scanner, mounted to the roof of the
seatainer, was controlled to compensate for pointing errors introduced by platform motion,
including those induced by ocean waves. The unique information obtained from this experiment
provided an opportunity for the first time to analyze the horizontal and vertical variability of marine
winds, offshore wind flow dynamics, and diurnal evolution of LLJ properties, and also to evaluate



model skill in an offshore setting, where high-quality wind measurements aloft are rare (Banta et
al. 2018; Djalalova et al. 2016; Pichugina et al. 2017a, 2017b).

Growing requirements for compact lidar configurations deployed on moving platform led
to the development of a new capability: a compact and robust microjoule-class pulsed Doppler lidar
system. Since 2018, the ARS/CSL group has focused on the development of such systems and
continuously updated design, measurement characteristics, and data acquisition techniques to
achieve the specific goals of each experiment.

The quantitative characteristics of wind and turbulence in the atmospheric layers occupied
by the wind turbine rotor blades (rotor layer) are crucial to wind energy, as is the information above
this layer to provide a meteorological context when considering profiles up to several hundreds of
meters above ground level (AGL). Furthermore, the region extending from the tops of the turbines
to the atmospheric boundary layer height plays a crucial role in the vertical entrainment of
momentum, which is an important driver of wind power capture (Meneveau, C. 2012;
Krishnamurthy et al. 2025).

Understanding the variability of winds across wind farms and under different conditions is
a critical factor in the planning and operation of wind projects. This goal can be achieved by
deploying a network of Doppler lidars over the wind farm or by taking measurements from a truck-
based mobile lidar. The accurate, motion-compensated measurements open an opportunity to
compare winds influenced by operational turbines (waked flow) with winds far from turbines (free
flow) along the driving path or to compare wind flows at different distances from turbine rows to
estimate the overall impact of the wind farm.

This paper aims to demonstrate the ability of truck-based Doppler lidar to provide high-
quality motion-compensated measurements in the boundary layer while driving around wind
turbines and present examples of analysis products obtained in August–September 2023 during the
multi-year American Wake Experiment (AWAKEN) campaign. Section 2 provides an overview of
ARS-developed mobile lidars, briefly describes technical parameters, motion-compensation, and
beam-stabilization systems, and discusses the lidar dataset. Section 3 presents the truck-based
mobile lidar, and discusses data obtained during an intensive operational period in Oklahoma.
Section 4 describes two case studies and provides analyses of the vertical, horizontal, and time-
evolving structures of wind flow in the presence of operating wind turbines for two selected days
characterized by differences in observed winds and boundary layer stability. Section 5 provides a



detailed analysis of the spatially and temporally varying structures of wind flow in the presence of
operating wind turbines for the two selected cases, showing wind speed and direction profiles at
various distances from turbines and comparing spatially distributed data from the mobile lidar with
data from nearby stationary Doppler lidars deployed in the research area. Section 6 contains
conclusions and recommendations.
**2. Development of the mobile micro-Doppler lidar system**
The compact micro-Doppler (MD) system deployment was achieved by a unique design of
a master oscillator power amplifier microjoule-class pulsed coherent Doppler lidar system in two
physically separated modules: the transceiver and the data acquisition system connected by an
umbilical cable (Schroeder et al. 2020). One module hosts the transceiver, which includes the
telescope, transmit/receive switch, and high-gain optical amplifier. The second module contains the
data acquisition system and several electro-optical components. This design, along with significant
decreases in the weight and the size of both modules, enables deployments of these systems on
small aircraft and pickup truck platforms that are otherwise inaccessible by commercial and
research instruments of similar design. The continuous updates and improvements of MD lidars
during the last several years led from version 1 (MD1) to version 3 (MD3). A detailed description
of versions MD1 and MD2, along with a short history of the development of stationary Doppler
scanning lidars in the NOAA/CSL ARS group, can be found in Schroeder et al. (2020).
Operation from a mobile platform faces many challenges, such as a constantly accelerating
reference frame and vibration while in motion. A significant obstacle to obtaining accurate wind
profiles from the high-precision lidar measurements using these techniques is compensating for the
pointing error and along-beam platform velocity due to platform motions. To address these issues,
the lidar is deployed with a *motion compensation* system that corrects the lidar velocity
measurement by estimating and removing the platform motion projected into the line-of-sight
velocity measurement in real time, and a *pointing stabilization* system that determines the platform
orientation and then actively stabilizes the orientation of the lidar beam in the world frame.
The development of the MD lidars and deployment of these compact systems on airborne,
shipborne, and truck-borne platforms (Figure 1) provided a new opportunity to study dynamic
processes in the atmospheric boundary layer in varied regions, from urban areas to remote locations





in complex terrain, and offshore. The flexible combination of temporal, vertical, and spatial
coverage of the study area provides a significant advantage over stationary profiling observations.
The MD3 design was optimized for operation from pickup trucks and ships. The small
modular footprint and weight of all subsystems allow their positioning in various compact spaces
and enable easy stabilization. The MD3 lidar system features two laser transmitters and two
channels to provide both continuous vertical-stare profiles of the vertical velocity $w$ and,
simultaneously, azimuth scans at 15° off zenith to give profiles of the horizontal wind speed and
direction using the velocity-azimuth display (VAD) technique (Browning and Wexler 1968; Banta
et al. 2002). The ability to do azimuthal scans at lower elevation angles, which can enhance
accuracy in the horizontal VAD wind estimate (see Banta et al. 2023), is currently under
development. The technical specifications of the MD3 lidar are given in Table 1.
Table 1. Typical specifications of the MD3 lidar

| Pulse Length | 30, 60, 90 m |
|---|---|
| Pulse repetition frequency | 20,000 Hz |
| Beam rate | 2–10 Hz |
| Pulse energy | 50 μJ |
| Beam diameter | 7.62 cm |
| Orientation | vertical |
| Maximum range | 7 km |
| Electrical power | 120 V, 30 A |
| Wavelength | 1.553 μm (invisible and eye safe) |

Many portable configurations of remote sensing instruments currently used for various
applications, including weather and atmospheric research, such as the Collaborative Lower
Atmospheric Mobile Profiling System (https://www.nssl.noaa.gov/tools/clamps), are considered
"mobile" systems. However, these systems must be delivered to the location of interest to provide
a stationary measurement or be used in a "go-and-stop-for-measurements" pattern. In contrast, the
mobile MD lidars developed at CSL/NOAA (Figure 1a–c) provide continuous measurements of $w$
and horizontal winds while the platform is moving, which is a significant advantage compared to



the constraint of stationary Doppler lidars to obtain vertical and horizontal wind profiles at one
location.

The truck-based measurements provide profiles of wind speed, wind direction, $w$, and aerosol

backscatter intensity, showing the wind flow variability in time, with height, and along the moving
path.

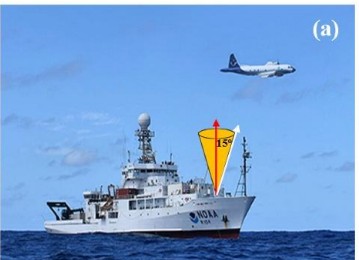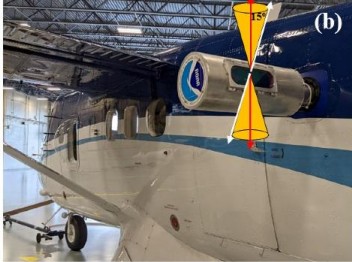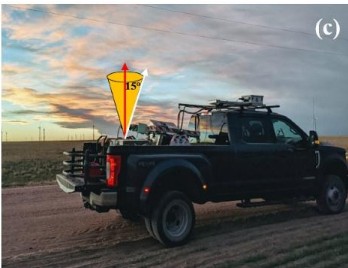


Figure 1. NOAA/CSL Mobile Doppler Lidar Systems: (a) Ship-based; (b) Aircraft-based; (c) Truck-based.

The mobile lidar measurements have been used for various environmental studies. The multi-

platform (aircraft and ground-based) setup was successfully used during recent wildfire and air-
quality experiments, providing a unique opportunity to characterize atmospheric processes,
including studies of fire plume transport dynamics, in better detail (Carroll et al. 2024; Strobach et
al. 2023, 2024). The combination of spatial and temporal coverage of the aircraft-based mobile lidar
measurements provides an advantage over traditional in situ or stationary profiling observations
offshore and inland, for example, to study the air quality of large urban areas
(https://csl.noaa.gov/projects/aeromma/cupids/). In the summer of 2024, the aircraft and truck-
based modifications were involved in multi-institutional projects to estimate emissions of methane,
greenhouse gases, and other significant air pollutants from oil and gas production facilities located
in urban and agricultural areas of Colorado (https://csl.noaa.gov/projects/airmaps/). Table A1 in
Appendix A shows a list of the CSL/NOAA field projects using mobile MD lidar on various
platforms. The results obtained from these experiments use the high precision and excellent
pointing accuracy of measurements from the ground-based, airborne, and shipborne deployments
and demonstrate success in developing a fully capable mobile Doppler lidar for environmental
studies.

WIND ENERGY SCIENCE DISCUSSIONS

eawe
european academy of wind energy

**2.1 Truck-based mobile lidar system**
The latest version of the truck-based lidar system (Figure 2), the PickUp-based Mobile
Atmospheric Sounder (PUMAS), was recently used to study the spatial structure of horizontal wind
and turbulence near wind farms in Colorado and north-central Oklahoma.

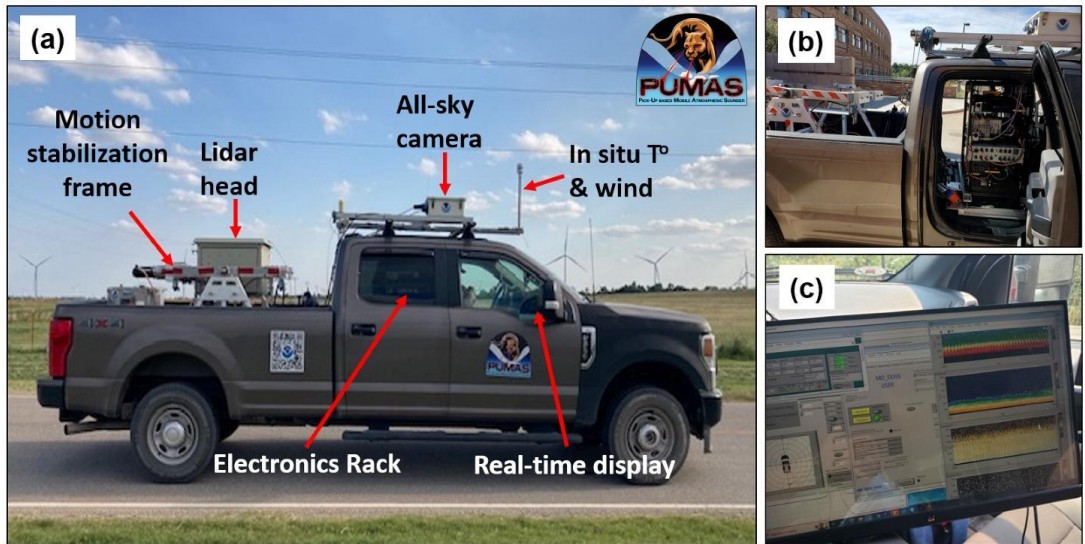


Figure 2. (a) Picture of PUMAS with indicated subsystems: Motion-stabilization frame, lidar head, all-sky
camera, the sensor for in situ measurements of temperature (Tº) and wind speed; (b) the electronics rack
located in the back of a cabin; (c) real-time display located in the front of the cabin.

The PUMAS system (Figure 2a) included a motion-stabilization frame, the MD3 lidar head,
an all-sky camera, and sensors for in situ temperature (T) and wind speed measurements. The
electronics rack is located in the back of the cabin (Figure 2b), and the real-time display is in the
front of the cabin (Figure 2c). PUMAS provided continuous motion-compensated measurements of
wind flow and turbulence profiles driving on highways and dirt roads within wind farms. The two
motion-stabilized lidar beams—vertically pointed and conically scanning with ±15º of zenith—
provided simultaneous profiles of horizontal wind vectors, aerosol backscatter intensity, and $w$
statistics from 60 m AGL to the top of the atmospheric boundary layer under normal atmospheric
conditions and absence of precipitation. Data were obtained with a temporal resolution of 1–4 Hz
and an along-beam resolution of 30 m. Wind speed profiles were obtained with an along-path
resolution of 300–600 m, and $w$ profiles every 10–30 m. Along-path resolution depends on the

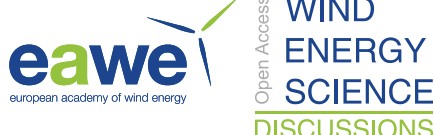

driving speed and the road conditions but can be modified by changing accumulation time or scan
settings in the real-time display software.
During several pilot studies, PUMAS was tested around wind farms in Colorado (Appendix
B, Figure B1) to obtain information on system performance, measurement errors, and driving
strategies. The analysis of data from these test drives helped to set the science goals and a
measurement strategy for the participation of PUMAS in the AWAKEN campaign.
**3. American Wake Experiment**
The AWAKEN campaign is a U.S. Department of Energy (DOE) project led by the National
Renewable Energy Laboratory (NREL). It is a multi-institutional, long-term (2021–2025) study in
the U.S. Great Plains aiming to understand the interaction between wind farms and their
surrounding environment and improve the performance of wake models. Wind farms in the north-
central Oklahoma study area are located over relatively flat terrain (Figure 3a). More information
on the AWAKEN goals can be found here: https://openei.org/wiki/AWAKEN. Participating
organizations deployed various in situ and remote-sensing instruments to the study area, including
14 stationary scanning Doppler lidars and seven wind-profiling lidars. The full description,
measurement objectives, and locations of the AWAKEN instrumentation can be found in the
overview paper (Moriarty et al. 2024). The first benchmark study within the International Energy
Agency Wind Task 57 framework focused on wind plant wakes (Bodini et al. 2024). Detailed
information on the coordinated measurements from in situ and remote-sensing instruments,
including turbine nacelle-mounted lidars, is provided in AWAKEN-related papers (Bodini et al.
2024; Debnath et al. 2022, 2023; Krishnamurthy et al. 2021, 2025; Letizia et al. 2023; Moriarty et
al. 2024). The long-term measurements from scanning lidars (Newsom, R.K. and Krishnamurthy
R, 2020) at the Atmospheric Radiation Measurement (ARM) Southern Great Plains (SGP) and
AWAKEN sites provide additional information on wind and turbulence in the surrounding area
(Moriarty et al. 2024).
To support the AWAKEN science objectives, the CSL/ARS team operated PUMAS to
provide motion-compensated measurements of 3D wind flow and turbulence profiles from 15 Aug
to 12 Sep 2023. The measurements were mainly taken within and around the King Plains wind farm
(Figure 3), which comprised 88 General Electric wind turbines with a rated capacity of 2.82 MW,
a hub height of 89 m, and a rotor diameter of 127 m.






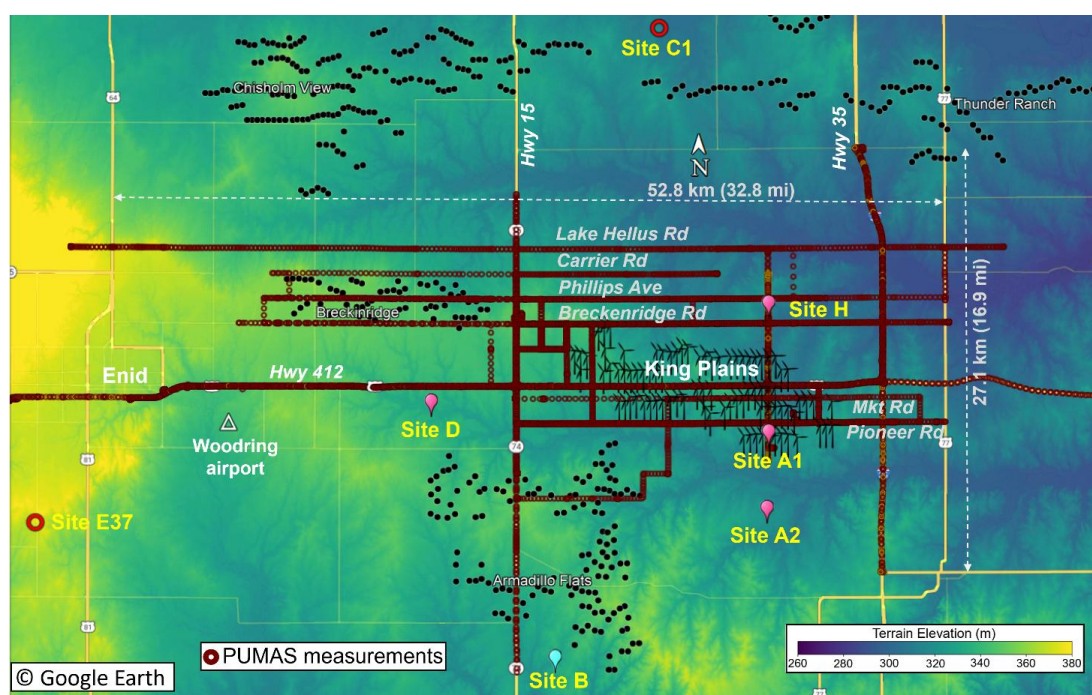


Figure 3. Wind farms in north-central Oklahoma are shown on the Terrain Elevation map (Debnath et al. 2022) by black dots. Turbine symbols show King Plains wind farm to underline the research focus on this area. Red circles indicate the ARM SGP highly-instrumented Central Facility C1 and the extended facility E37 (https://www.arm.gov/capabilities/observatories/sgp). Pink and cyan pins indicate AWAKEN lidar and ASSIST sites used in this paper. The roads (transects), covered by PUMAS during AWAKEN, are shown by dark red circles where each circle represents a profile measurement from 64 m up to several kilometers AGL. The white triangle indicates the Woodring Regional Airport in Oklahoma, located about 8 km southeast of the central business district of Enid, Oklahoma.

By considering the predominant wind direction estimated from various model forecasts at the Enid Woodring Regional Airport in Oklahoma, a driving plan for each day was designed to sample waked and free flows at various distances from the wind turbines (Figure 3). Transects were repeated several times during 5–6 hours of measurements each day. At the beginning and end of each transect, 5-minute measurements were made in a stationary position, and these data were used to evaluate the system performance, as shown in Sect. 3.3.

In addition to PUMAS measurements, data from stationary Doppler lidars deployed at various AWAKEN sites (Figure 3) were used for this paper. Data from the PNLL flux station were used to estimate near-surface stability. Temperature and water vapor mixing ratios were estimated



through the TROPoe retrieval (Turner and Blumberg, 2018; Turner and Loehnert, 2014) based on
observations from the NREL ASSIST-II spectroradiometer (Michaud-Belleau et al. 2025)
measurements at Site B (Figure 3). The list of instruments used in the paper is given in Table 2.
Table 2. Coordinates of sites and types of instruments used in the paper.

| Site | Latitude | Longitude | | Instrument |
|---|---|---|---|---|
| **PUMAS** | varied | varied | | NOAA/CSL motion compensated system with HALO XR lidar |
| **H** | 36.4370 | -97.4077 | sh.lidar.z02.c1 | AWAKEN scanning Doppler lidar HALO XR |
| **A1** | 36.3623 | -97.4078 | sa1.lidar.z03.c1 | AWAKEN scanning Doppler lidar HALO XR |
| **A2** | 36.3182 | -97.4090 | sa2.lidar.z01.c1 | AWAKEN scanning Doppler lidar HALO XR |
| | | | sb.met.z01.b0 | PNNL flux station |
| **D** | 36.3799 | -97.6465 | sd.lidar.z01 | Fraunhofer IWES's WindCube v2.0 |
| **C1** | 36.6050 | -97.4850 | sgpdlprofwind4newsC1.c1 | ARM scanning Doppler lidar HALO XR |
| **E37** | 36.3110 | -97.9280 | sgpdlprofwind4newsE37.c1 | ARM scanning Doppler lidar HALO XR |
| **B** | 36.2316 | -97.5587 | sb.assist.z01.c0 | Assist II-11 |

**3.1 Meteorological conditions during PUMAS measurements in northern Oklahoma**
According to the Oklahoma Climatological Survey (https://www.ou.edu/ocs/oklahoma-
climate), the AWAKEN study area is in the North Central climate division. This northern section
of the state is less influenced by the warm, moist air moving northward from the Gulf and
experiences less cloudiness and precipitation compared to the southern and eastern portions of the
state. Still, summers there are long and usually quite hot.
The surface wind statistics at the Enid Woodring Regional Airport, located 6.4 km southeast
of downtown Enid, show predominant south-southeast wind directions in August and September
2023 and 5 m s$^{-1}$ mean winds with occasional gusts up to 10 m s$^{-1}$. The frequency of weak (1–4 m





s$^{-1}$) winds is high for both months (71% in August and 63% in September), whereas stronger winds
(4–11 m s$^{-1}$) were less common (17% in August and 25% in September). The August–September
2023 average temperature in Enid was 86–93 °F (30–34 °C) for the daytime and 68–73 °F (20–23
°C) for nighttime, with 20–22 sunny days each month and two rainy days on 13–14 September
(www.windfinder.com/windstatistics/enid_woodring_regional_airport).

The ARM SGP atmospheric observatory with various in situ and remote-sensing instrument

clusters located in north-central Oklahoma and south Kansas near the AWAKEN study area (Figure
3). The scanning Doppler HALO Photonics lidars provide long-term wind and turbulence
measurements (Newsom R. K. and Krishnamurthy R. 2020) at the SGP central facility, C1, and
four extended sites (E32, E37, E39, E41) and are used in many studies and experiments such as the
Plains Elevated Convection at Night field campaign (Geerts et al. 2017) or the Land-Atmosphere
Feedback Experiment (Wulfmeyer et al. 2018; Pichugina et al. 2023, 2024).

A 6-year analysis of (2013–2019) Doppler lidar data at C1 located north of the King Plains

wind farm (Figure 3) confirms predominant southeast and south-southeast wind directions at 91 m
AGL in August and September (Krishnamurthy et al. 2021). Another detailed study of winds from
Doppler lidars at the five SGP sites revealed that the interannual (2016–2022) variability of monthly
mean summer nighttime winds in the layer of 700 m AGL was more significant (4 m s$^{-1}$) compared
to the wind variability (1–3 m s$^{-1}$) between sites, which are separated by 56–77 km, characterized
by different vegetation types, and have elevations that vary between 279 and 379 m above sea level
(ASL) (Pichugina et al. 2023). They also reported predominant south-southeast nighttime winds at
all sites and frequent wind maxima at ~300 m.

Wind roses of 91 m winds from stationary Doppler lidar measurements on Aug. 15–Sept.

12, 2023, at two ARM SGP sites (C1 and E37) closest to the King Plains wind farm show wind
directions from north to southwest with predominant southeasterly winds (Figure 4a, b). Time-
height cross sections of winds averaged over 15 Aug–12 Sep 2023 (Figure 4c, d) were moderate
(8–12 m s$^{-1}$) at night and weaker (4–6 m s$^{-1}$) during daytime. At both sites, wind directions below
300 m were primarily southeasterly, with some episodes of southerly winds at higher elevations.
At C1 (Figure 5c), LLJ development is evident within 200–700 m AGL around ~0500–1200 UTC,
whereas at the western E37 site, located 51 km to the southwest of C1 (Figure 3), the LLJ developed
earlier in the 100–700 m layer.

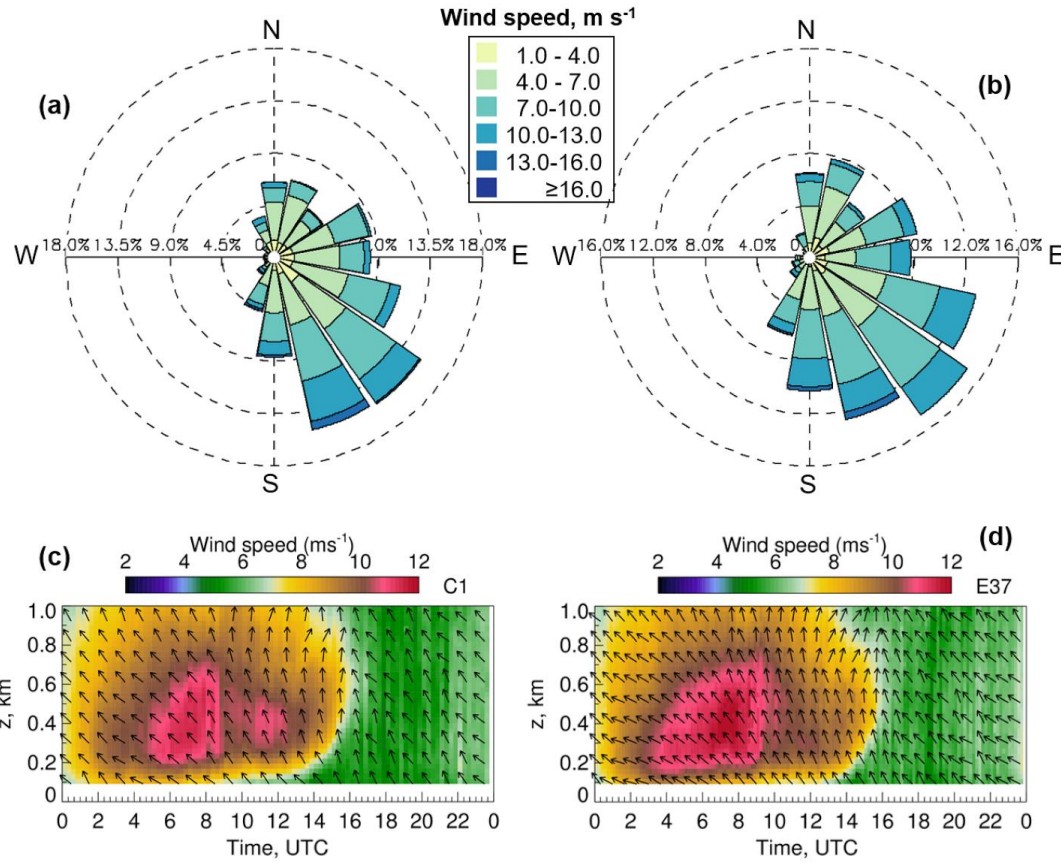

Figure 4. Wind roses of 91 m winds from Doppler lidars at the ARM SGP sites (a) C1 and (b) E37 for all hours of measurements during 15 Aug-12 Sep 2023. (c, d) Time-height cross sections of period-mean wind speed (colors) and wind direction (arrows) from each Doppler lidar. Local Time=UTC-5.

**3.2 Statistics of PUMAS measurements.**

As mentioned, PUMAS participated in the AWAKEN experiment from 15 Aug to 12 Sep 2023. Only 20 days of good measurements were available due to poor weather conditions (heavy rain) and technical issues such as flat tires or lidar-system-component issues. Four days were spent on a round trip between Boulder, Colorado, and Enid, Oklahoma. During each 982 km one-way commute, PUMAS provided continuous measurements of wind speed, wind direction, and $w$. The system performance was monitored and corrected as needed in real time, including motion-compensation parameters such as transceiver pitch, roll, and heading; platform velocity and coordinates; and estimates of the lidar beam azimuth and elevation in a world reference frame.



Overall, during the 20 driving days, PUMAS was on the road 81 hours, covering 3930 km
(2443 mi) and providing 16,955 profiles of horizontal winds and *w* excluding data obtained during
Denver–Oklahoma commutes.
The distribution of PUMAS operation hours (Figure 5a) shows that the most intense
measurement period was in the late morning to midday (1500–2000 UTC). Nighttime
measurements during stable conditions, when turbine wakes could be better observed due to the
more substantial wind speeds and lower turbulence, were limited by the country road conditions
and pure visibility of the upcoming crossroads traffic. It was expected that some events, such as the
nocturnal LLJ, a frequent Great Plains phenomenon (Banta et al. 2002), would not be captured in
the late mornings. However, the dissipation times of the LLJ often depend on synoptic conditions,
and in some cases, LLJ can be observed after sunrise hours (Carroll et al. 2019; Squitieri B. J. and
W. A. Gallus 2016; Pichugina et al. 2023).
The wind rose of the 64–160 m layer wind speeds (Figure 5b) shows the dominance of
southeasterly winds during PUMAS measurements. Strong (>15 m s$^{-1}$) winds were observed in
13% of the southerly cases, followed by 10% in southeasterly and 7% in southwesterly directions.
Based on the *dθ/dz* data from the ASSIST at Site B, the majority of PUMAS measurements were
taken under unstable conditions (88.3%) as estimated from the ASSIST measurements at Site B.
Stable conditions were observed in 7.8% of cases, and near-neutral conditions were observed in
3.9% of cases.



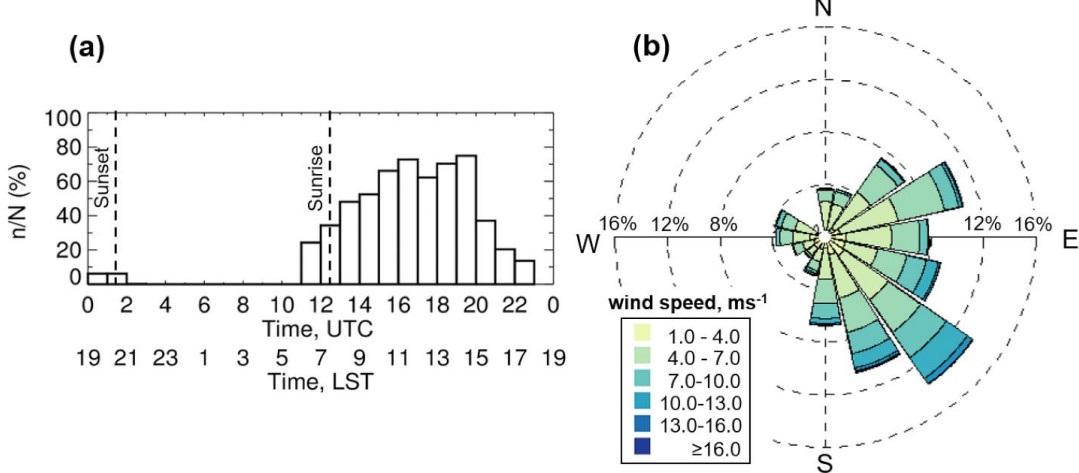


Figure 5. (a) Diurnal distribution of the PUMAS hours of operation during AWAKEN; (b) Wind rose of
turbine level (64–160 m) winds from the PUMAS measurements.

## 3.3 Platform stabilization and motion correction

Active stabilization and pointing correction, implemented in the mobile lidar system, compensates for truck motions such as pitch and roll (Figure 6a, b) removing the effect of bumps on $w$ while PUMAS is moving. In other words, the stabilization and motion-corrected system allow measurements of the $w$ to be obtained without mixing in the projection of the horizontal wind speeds and their variation. Correction of the pitch and roll motions keeps the lidar beam elevation angle in a world frame at 89.21° on average with a standard deviation of ±0.96 (Figure 6c) to obtain corrected line-of-sight velocity with an accuracy of -0.04 ± 0.31 m s$^{-1}$. An example of the motion-corrected vertical velocity from PUMAS measurements on 7 Sep (Figure 6e) shows significant turbulence in the first 1 km ASL and illustrates the 287–415 m variability of the terrain covered by PUMAS on this day. The mean difference between measured and motion-corrected $w$ at 105 m (Figure 6f) is 0.08 ± 0.32 m s$^{-1}$.

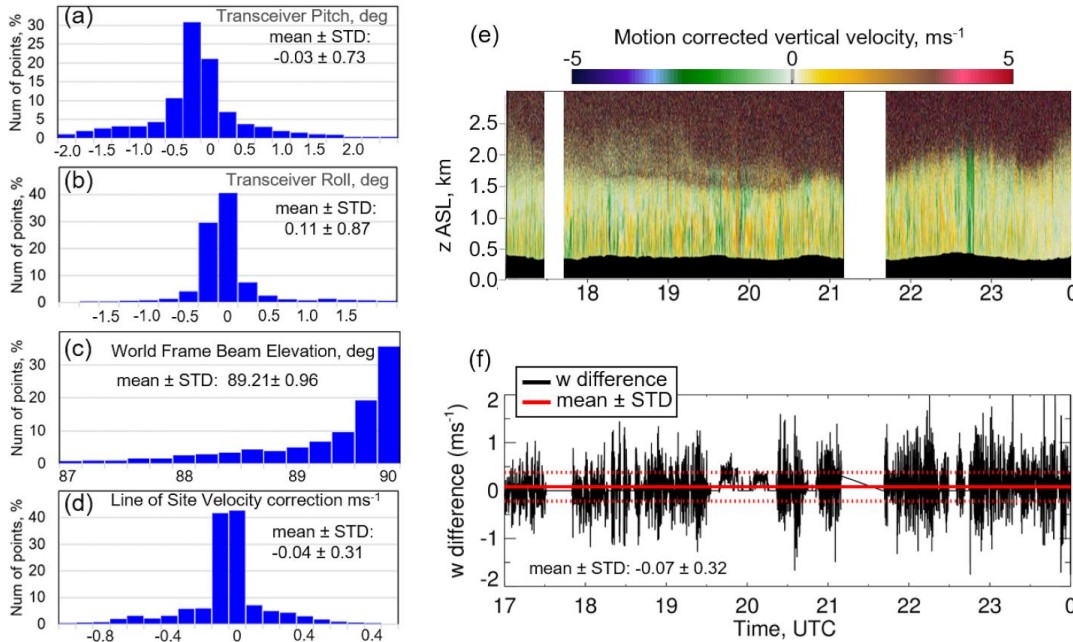

Figure 6. (a–d) Distributions (%) of the truck motion correction from PUMAS vertical velocity measurements on 15 Aug–12 Sep 2023, during AWAKEN. Mean ± standard deviation (STD) is shown on the panel for each parameter. (e) A sample of motion-corrected vertical velocity measurements from 17:00 to 24:07 UTC on 7 Sep 2023. Terrain elevation above sea level (ASL) covered by PUMAS on this day is shown in black. The white areas indicate missing data. (f) Time series of a (black) difference between measured and motion-corrected vertical velocity on 7 Sep 2023 at 105 m above ground level (AGL). Red solid line shows a period-mean difference. Dotted red lines show STD from the mean.

As mentioned, PUMAS provided 5–7 min of measurements in a stationary position at the beginning and the end of each transect. Measurements collected by PUMAS in a stationary position or while driving within a 2 km radius of a DOE stationary lidar at Site A1 or H are used to estimate the accuracy of PUMAS's horizontal wind speed and direction by comparing the PUMAS and DOE lidar measurements as shown in Figure 7 and summarized in Table 3. The different number of wind speed and direction points (count) for each case is because the 3-sigma outlier rejection (see Pichugina et al. 2020) to the 1:1 fit was applied for speed and direction separately, leading to a different number of outlier points removed for speed and for direction. High correlation coefficients were obtained for wind speed (0.83–0.96) and wind direction (0.93–0.99) from PUMAS measurements in a stationary position and while moving except two cases when correlation coefficients were 0.65 between wind speed from the stationary PUMAS and Doppler lidar at Site A1, and 0.62 between wind direction from the moving PUMAS and Doppler lidar at Site H. The



larger offset in wind direction histograms was observed between PUMAS and Doppler lidar at Site
H. Detailed analysis of these results is beyond the scope of this paper.

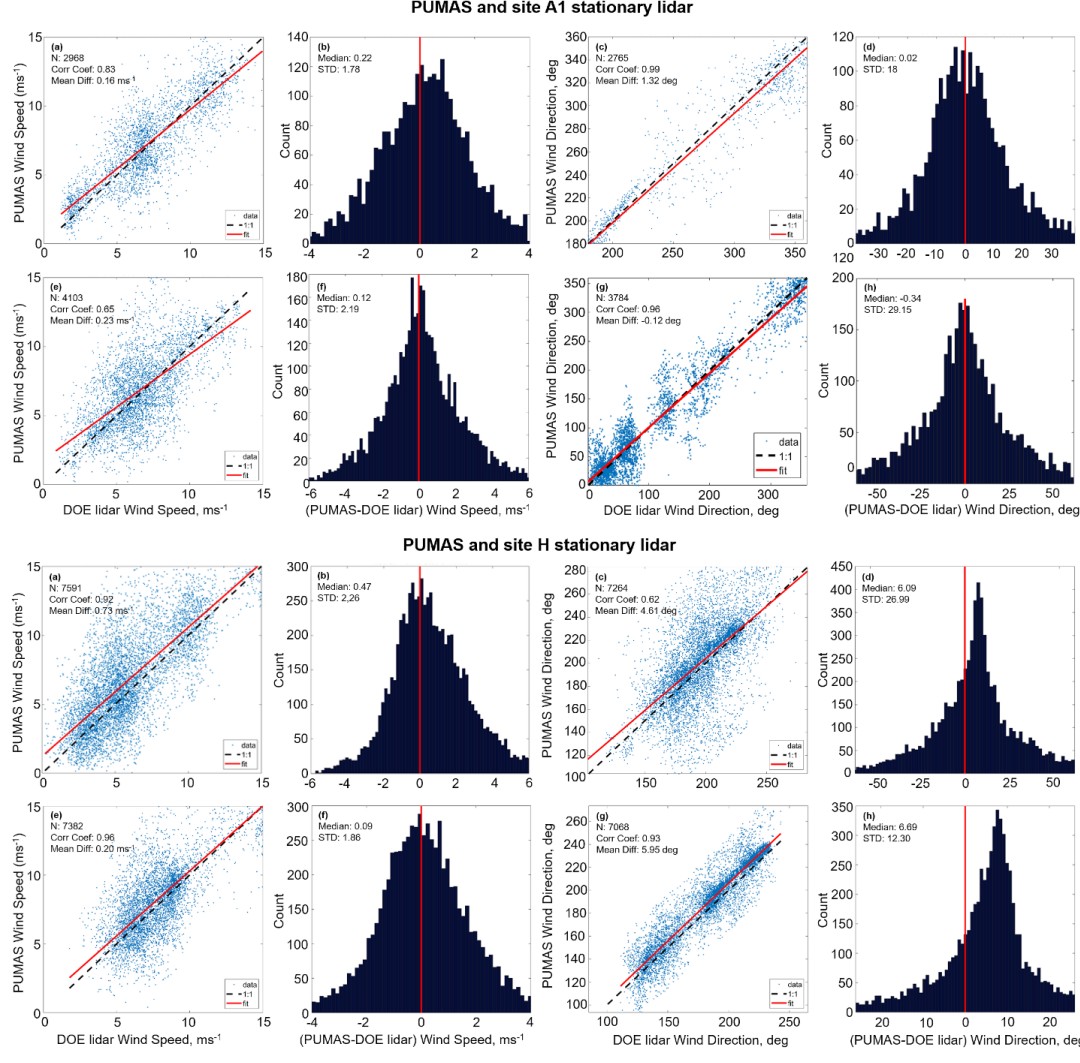

Figure 7. Comparison of horizontal wind and direction between PUMAS and DOE stationary Doppler lidar
at Sites A1 and H: (a–d) from PUMAS measurements in stationary position collected within 2 km radius
from DOE stationary Doppler lidar; (e–h) from moving PUMAS measurements collected within 2 km radius
from DOE stationary Doppler lidar.






Table 3. Statistics from the comparison of wind speed and direction measurements from PUMAS
and stationary Doppler lidars.

| PUMAS vs. Stationary Doppler Lidar at Site A1 | | | | | | |
|---|---|---|---|---|---|---|
| **PUMAS measurements** | ***Data*** | ***Scatter plots statistics*** | | | ***Histogram statistics*** | |
| | | *Count* | *Cor Coef* | *STD* | *Medium* | *STD* |
| **Stationary** | ***Wind speed*** | 2968 | 0.83 | 0.16 | 0.22 | 1.78 |
| | ***Wind direction*** | 2765 | 0.99 | 1.32 | 0.02 | 18.0 |
| **While moving** | ***Wind speed*** | 4103 | *0.65* | 0.23 | 0.12 | 2.19 |
| | ***Wind direction*** | 3784 | 0.96 | -0.12 | -0.34 | 29.15 |
| **PUMAS vs. Stationary Doppler Lidar at Site H** | | | | | | |
| **Stationary** | ***Wind speed*** | 7591 | 0.92 | 0.73 | 0.47 | 2.26 |
| | ***Wind direction*** | 7264 | *0.62* | 4.61 | 6.09 | 26.99 |
| **While moving** | ***Wind speed*** | 7382 | 0.96 | 0.20 | 0.09 | 1.86 |
| | ***Wind direction*** | 7068 | 0.93 | 5.95 | 6.69 | 12.30 |


Overall, Figures 6 and 7 and Table 3 clearly illustrate success in developing a fully capable
mobile Doppler lidar that compensated for the truck's motions to provide accurate wind
measurements. The uncertainty of the horizontal wind speed and direction estimated by the VAD
technique (Banta et al. 2013) from PUMAS line-of-sight velocity measurements during AWAKEN
was found to be very small with mean and standard deviations of $0.014 \pm 0.008$ m s$^{-1}$ for wind
speed and $0.12° \pm 0.18°$ for wind direction. The accuracy of motion-compensated measurements
from mobile lidars was tested against stationary Doppler lidar measurements during several field
campaigns. Examples of active stabilization and the accuracy of diurnal measurements from ship-
based lidar during the offshore VOCALS campaign (Table A1) are provided in the Supplemental
Material (S1a, b). Examples (S2a, b) illustrate a high correlation for wind speed (0.89, 0.90) and
direction (0.93, 0.99) obtained from two experiments while PUMAS was driving within a 2.5 km
radius from the stationary lidar (S2c) and when PUMAS provided measurements in a stationary
position for several months (S2d).





**4. Selected case studies: 5 and 7 September.**

Two days, 5 and 7 Sep, were selected to illustrate the PUMAS measurements and analysis techniques. The data on these days were obtained during morning transition (5 Sep) and day-evening transition (7 Sep) periods, characterized by some difference in wind conditions and BL stability. Figure 8 shows wind speed (Figure 8a, c) and direction (Figure 8b, d) on these days from stationary Doppler lidars at SGP Site C1 (left) and SGP Site E37 (right).

**4.1 Wind speed and direction from stationary Doppler lidars**

On 5 Sep (Figure 8a, b), during a period of PUMAS operations in the early morning hours (1143–1645 UTC, 0543–1045 LST), both SGP lidars show strong (15–25 m s$^{-1}$) wind speeds and the development of the LLJ at 0200–1500 UTC (LST=UTC-5hours) with the LLJ maximum at 600 m. Wind directions (Figure 8c, d) in the first 200–300 m AGL changed from southeasterly at nighttime, veering to southwesterly from late morning to afternoon and becoming northerly in the evening hours (after 1800 UTC). The wind speed ramp-down event observed at ~0900–1100 UTC below 400 m, is most likely another example of an atmospheric bore, as analyzed in this region by Pichugina et al. (2024). It corresponds to a transient shift to a more southwesterly wind direction. Such significant increases or decreases in wind speed lasting for a half-hour or more are difficult to forecast but may significantly affect turbine operations.

On 7 Sep (Figure 8e–h), both SGP lidars showed weak (<4 m s$^{-1}$) nighttime winds that increased to 8–12 m s$^{-1}$ by 0900–1000 UTC (Figure 8c). The LLJ of ≥15 m s$^{-1}$ developed at Site C1 at 1400–1500 UTC below 400 m while stronger (15–20 m s$^{-1}$) LLJ developed at Site E37 around 1300–1500 UTC below 300 m. Wind directions (Figure 8g, h) were primarily east-southeasterly (100°–150°) at both sites.





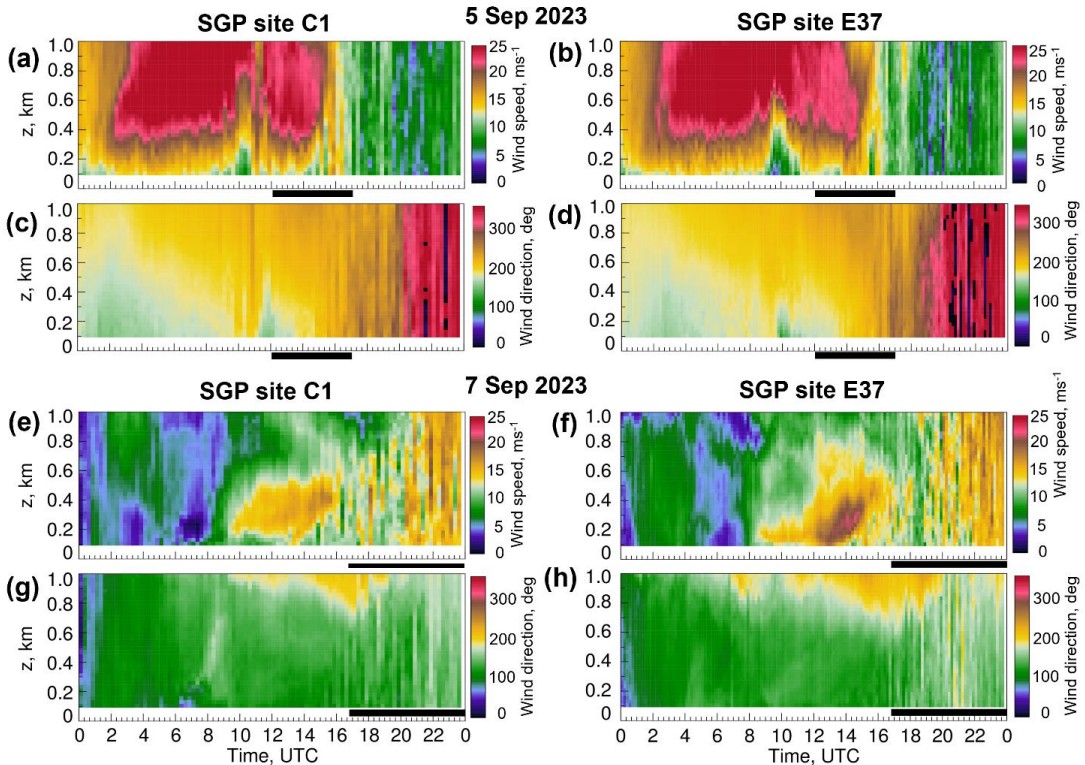

Figure 8. Time-height cross sections of wind speed and wind direction from stationary lidar measurements at the SGP Sites C1 (left) and E37 (right) measurements on (a–d) 5 Sep and (e–h) 7 Sep 2023. Black lines indicate periods of PUMAS measurements on these days. The temporal resolution of lidar data at C1 is 15 min and at E37 is 10 min. Lidar data at SGP sites can be found at the DOE ARM archive: http://dx.doi.org/10.5439/1178582.

Time series of wind speed and direction (Figure 9) at the six lowest heights from all stationary lidars depicted in Figure 4 also show similar trends in the evolution of wind flows, despite a significant distance between these instruments and locations at different terrain over the AWAKEN research area (Figure 4). In Figure 9, all lidars show highly variable wind speeds on 5 Sep, with an indication of a ramp event around 0900–1200 UTC and weaker, less variable winds on 7 Sep.



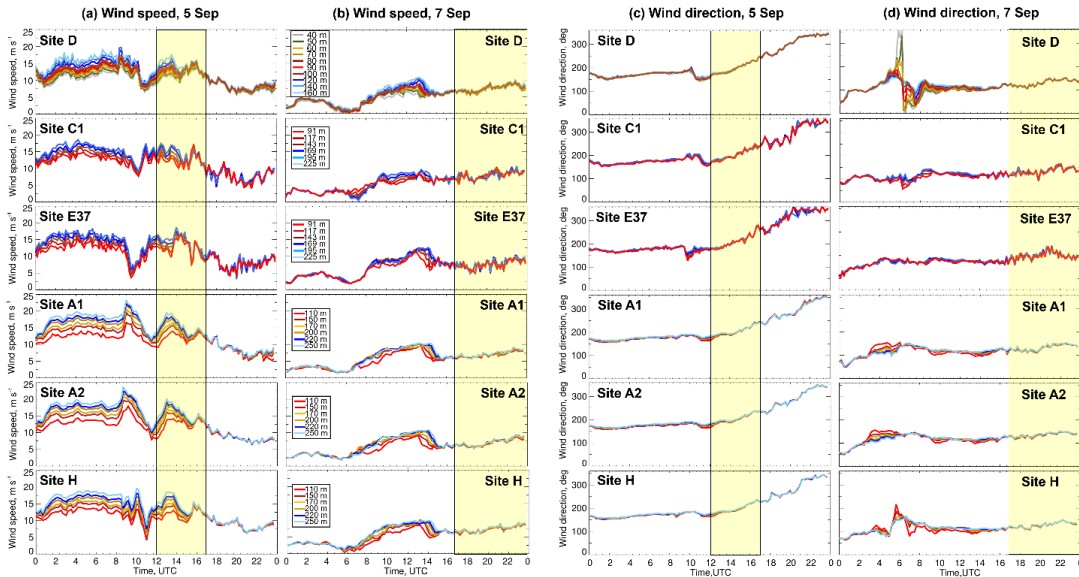

Figure 9. Time series of (a, b) wind speed and (c, d) wind direction from six stationary Doppler lidars at lowest heights on 5 Sep and 7 Sep The location of lidar sites (Site D, Site C1, Site E37, Site A1, Site A2, and Site H) are shown in Figure 4. The heights of measurements are indicated in the legend for each lidar. Periods of PUMAS operations in a field on 5 Sep and 7 Sep are highlighted by the yellow color.

Interestingly, this pattern changed little between lidar measurements of inflow at Site A2 and waked flow at Site H during the period of PUMAS measurements highlighted by the yellow color (Figure 9). On 5 Sep, the mean wind speed at Site A2 was 0.8 m s$^{-1}$ larger, and on 7 Sep, mean winds were 0.46 m s$^{-1}$ weaker compared to Site H. The difference in wind direction between sites was 6.39° on 5 Sep and 7.83° on 7 Sep.

### 4.2. Stability on 5 and 7 September 2023

The virtual potential temperature ($\theta v$) computed from the TROPoe retrievals (Turner et al. 2014) of temperature and water vapor mixing ratio from thermodynamic profiler (ASSIST) data at Site B is shown (Figure 10a, b) for 5 Sep and 7 Sep. Stability estimates based on the virtual potential temperature gradient ($d\theta v/dz$) show stable conditions at the beginning of PUMAS measurements on 5 Sep that changed to unstable by the end of the period (Figure 10c), whereas on 7 Sep, the unstable conditions were observed during all hours of PUMAS operations (Figure 10d). Sonic anemometer data from the PNNL flux station at Site A2 confirms the differences in stability between measurement hours on the two days (Figure 10e, f). Stability based on the Obukhov length

(L) threshold (Krishnamurthy et al. 2021) was stable at the beginning of PUMAS measurements on
5 Sep and changed to neutral in the middle and unstable at the end of this period (Figure 10e). On
7 Sep, very unstable conditions were observed at the beginning of the period changing to less
unstable conditions by the end (Figure 10f).

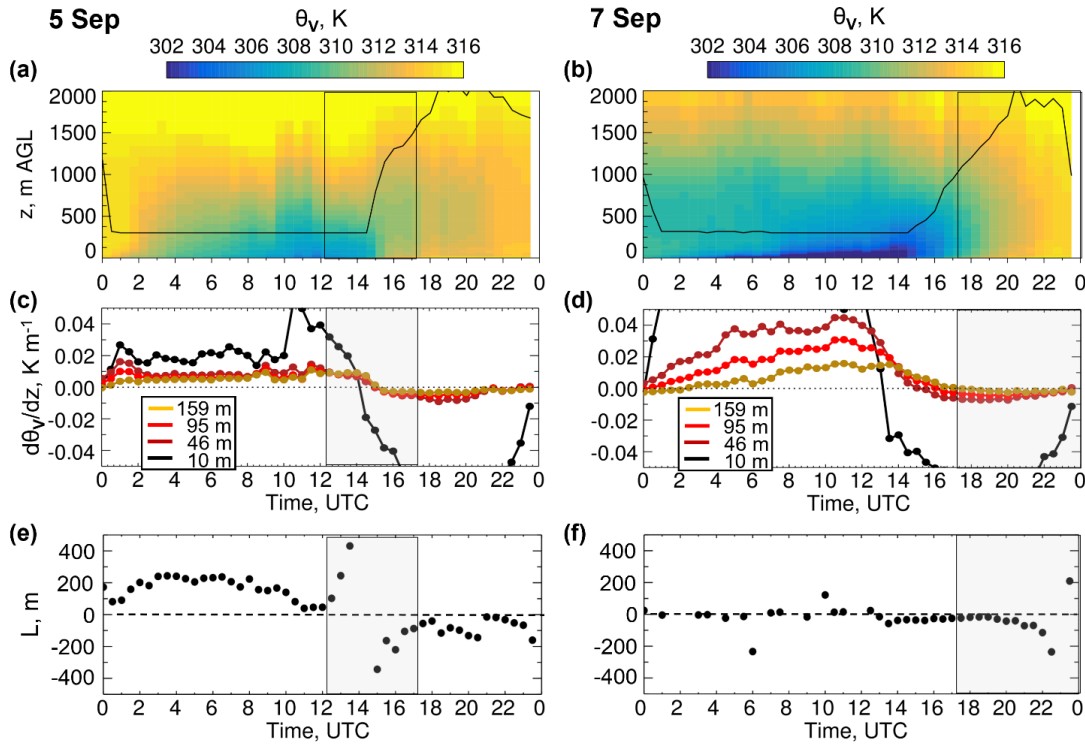

Figure 10. (a) Virtual potential temperature ($\theta v$) from ASSIST data at Site B on (a) 5 Sep and (b) 7 Sep.
Black lines show planetary boundary layer height (m) derived from the retrieved fields. Virtual potential
temperature gradient ($d\theta v/dz$) on (c) 5 Sep and (d) 7 Sep at 10 m AGL and three heights within the limits
of turbine blades. (e, f) Obukhov length from the PNNL flux station at Site A2 for these days. Gray shaded
areas indicate periods of PUMAS measurement on each day.
**5. PUMAS measurements on 5 and 7 September 2023**

The PUMAS data, obtained with high temporal resolution and a significant spatial

distribution over driving transects (see the following subsections), show a similar evolution of wind
speed and direction to the stationary SGP lidars (Figure 8) for the period of PUMAS operations.



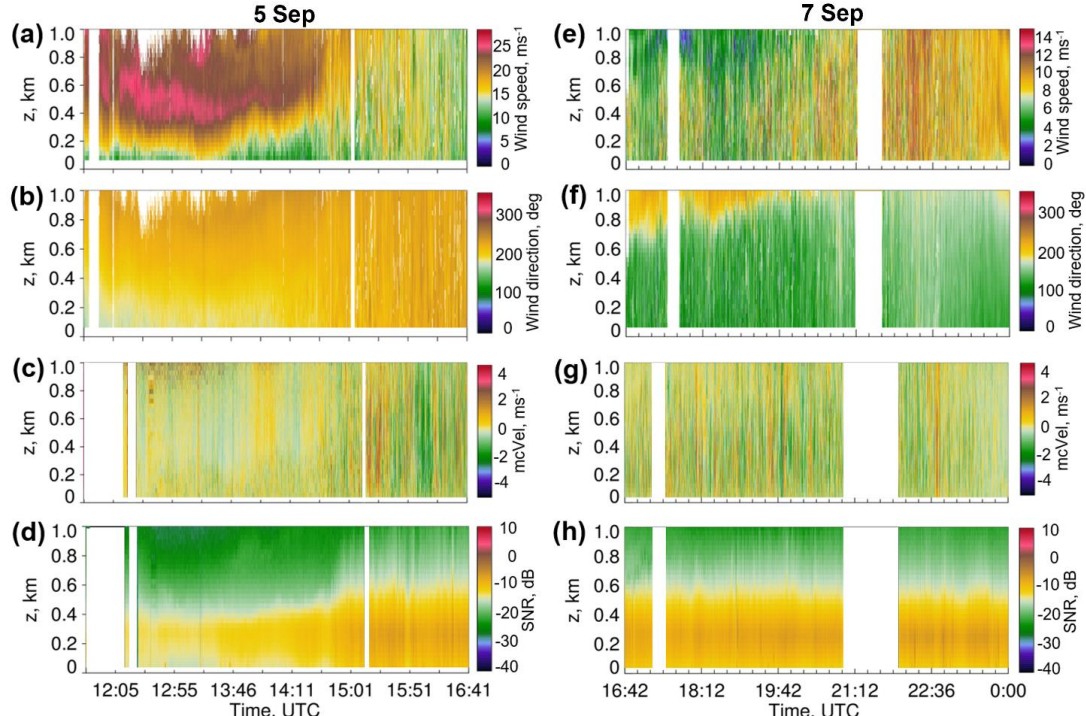

Figure 11. PUMAS-measured time-height cross sections of (a) wind speed, (b) direction, (c) motion-corrected vertical velocity, and (d) SNR (signal-to-noise ratio) intensity from simultaneous (Figure 11a, b, e, f) scanning and (Figure 11c, d, g, h) vertically-pointing data on 5 Sep (left column) and 7 Sep (right column). White areas indicate missing data.

On 5 Sep (Figure 11a–d), PUMAS measurements in the morning hours (1143–1645 UTC) show an LLJ mixing out after 1500 UTC. The data captured strong (≥15 m s⁻¹) morning (~1200–1500 UTC) wind speeds at higher elevations and the LLJ of ~25 m s⁻¹ at 500–600 m (Figure 11a). The wind directions were predominantly south-southwesterly (~200°) with short periods of southerly winds below 200 m (Figure 11b). Stronger convective mixing was observed after 1500 UTC (Figure 11c) as BL depth increased from 400 m to 600 m AGL (Figure 11d) and stability within rotor heights changed from stable to unstable (Figure 10c).

On 7 Sep (Figure 11e–h), PUMAS operated in the field for about 7 hours from late morning to the evening (1642–0007 UTC). Similar to 5 Sep, the agreement in trend (wind speeds increasing through the period) between data from stationary SGP lidars and PUMAS measurements was evident, although PUMAS sampled somewhat weaker winds. The daytime (1642–2100 UTC), southeasterly (120°–140°) winds of 5–8 m s⁻¹ increased by the evening to 10–12 m s⁻¹ (Figure 11e),





and veered to south-southeasterly (160°–170°) below 600 m (Figure 11f). The steady mixing with
the BL height to >600 m was observed during most of a period (Figure 11g, h) characterized by the
unstable BL conditions (Figure 10d).
The next sections will provide a closer look at PUMAS measurements during selected
days starting with 7 Sep, the longest period of measurements characterized by moderate (6–12 m
s$^{-1}$) wind speed and unstable BL conditions, which were common for most days during PUMAS
operations.

## 5.1 7 September case study, southeasterly winds

Throughout the previous sections, 5 Sep was discussed first, then 7 Sep. Here, we change
the order and start with the case study on 7 Sep, as it was the longest period of PUMAS
measurements, and these data were taken during the most frequent (Figure 5a) late-morning (1600–
2000 UTC) hours. Relatively calm wind speeds and southeasterly directions this day are more
common for many other days in contrast to the 5 Sep case of strong, southwesterly winds. On 7 Se,
PUMAS operated in the field for about 7 hours and 25 min (1642–0007 UTC), covering more than
422 km. A 3D visualization of wind profiles (Figure 12) measured on 7 Sep along several transects,
out of 34 total for the day, illustrates stronger (≥10 m s$^{-1}$) winds in green colors compared to weaker
(≤5 m s$^{-1}$) winds shown by purple colors.

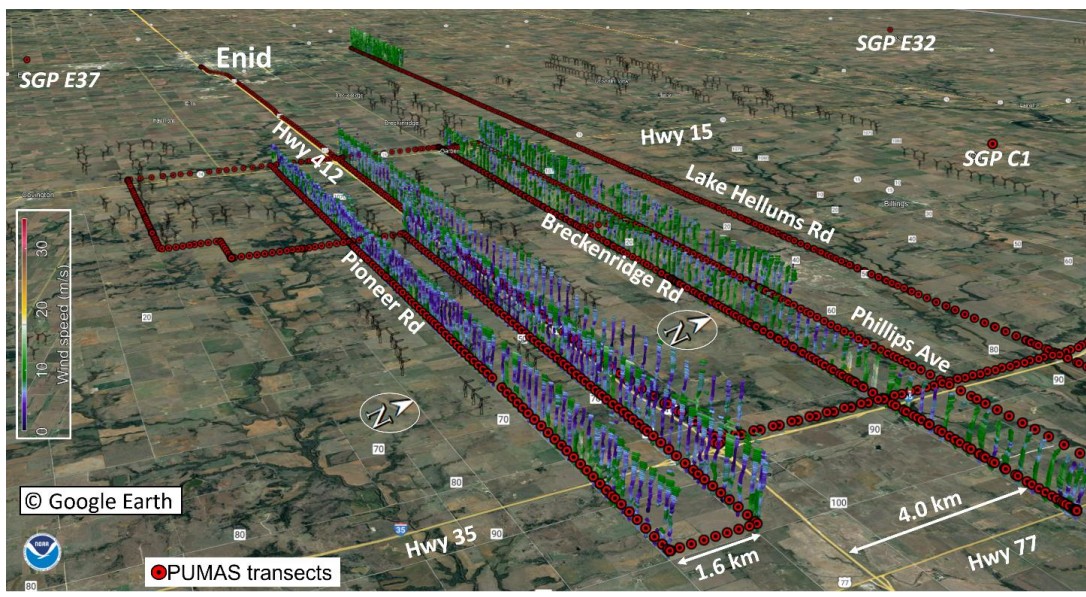





Figure 12. Samples of wind profiles along some transects on 5 Sep 2023, embedded on Google Earth, are
rotated clockwise ~45° for a better view. Profiles are shown up to 1.5 km AGL, and wind speed is scaled
from 0 to 30 m s$^{-1}$ according to the color scale on the left side of this figure. The horizontal distance between
profiles is about 300 m. White arrows indicate distances between illustrated transects along the named roads.
Gray circles indicate the ARM SGP sites (C1, E37, and E32).
**5.2 Technique to estimate free and waked flows**
A technique to estimate wind speed for sections of a transect that are in the shadow of wind
turbines (waked flow) or free from the turbine influence (free flow) is based on the density of
upstream wind turbines that may impact wind measurements, computed within 10 km from the road
(Figure 13b, e) including all Breckinridge wind farm turbines located within 2–4.7 km from this
road (note a slight spelling difference in the road and wind-farm names). This example did not
consider some of the King Plains and all Armadillo Flats turbines located more than 10 km from
the road. The influence of turbines on wind-speed measurements (turbine shadow) was estimated
within a 20° arc (±10° turbine shadow) from each point of a PUMAS measurement of wind
direction. Sections of a transect indicated by red in the wind time series (Figure 13a, d) are
considered waked, whereas those considered as not influenced by wind turbines (free flow) are
blue.
Figure 13 a, d shows time series of the rotor-layer (64–150 m) mean wind speed measured
during the east-to-west (EW, 59.2 km) and west-to-east (WE, 55.4 km) transects on Lake Hellums
Rd. (Figure 12). Mean rotor-layer winds in the free-flow sectors along the EW transect increased
from 9.1 to 9.9 m s$^{-1}$, whereas on the return WE transect, the winds decreased from 8.9 to 8.2 m s$^{-1}$
$^{-1}$. The free-flow wind speeds were thus stronger for the western sector by 0.7-0.8 m s$^{-1}$, most likely
due to terrain differences, and the winds slowed by ~1 m s$^{-1}$ in the time between the two sampling
legs.   Significant spatial variation of the wind speed within both the waked flow and the free flow
sectors reflects the significant natural atmospheric variability characteristic of this midday
convective period and appears larger than the mean speed differences between waked and free flow
regions.



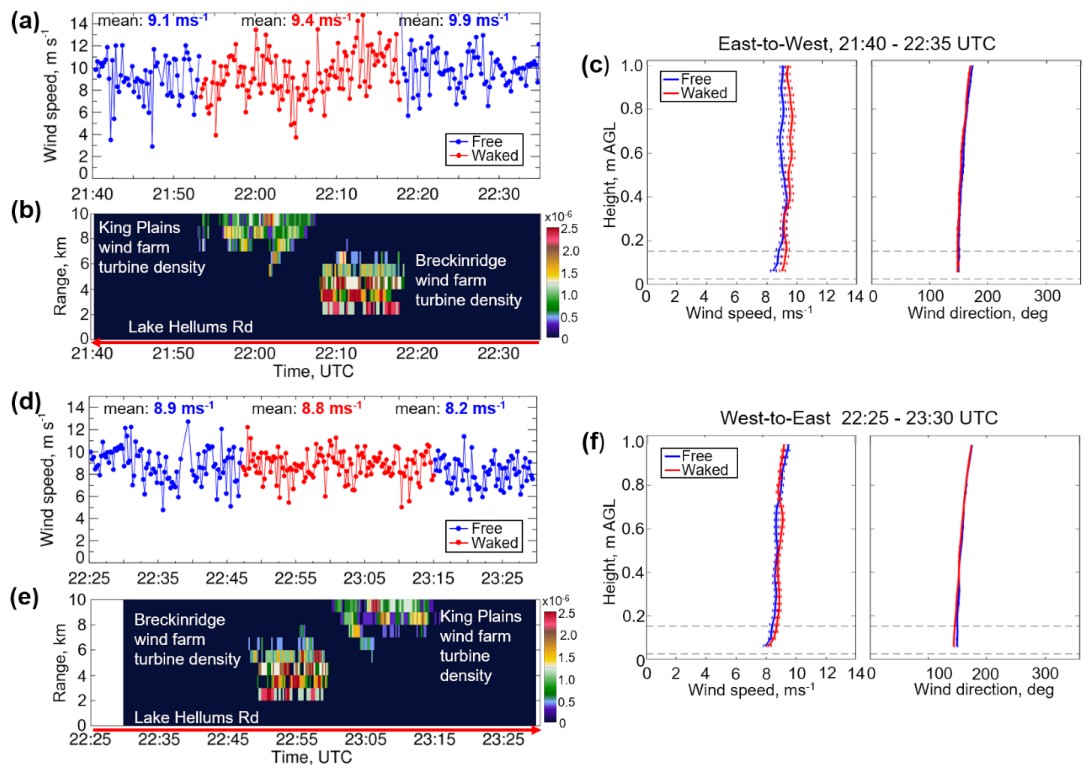


Figure 13. Time series of wind speed averaged over the rotor-layer (64–159 m) height from PUMAS
measurements on Lake Hellums Rd. during (a) east-west (49.3 min) and (b) west-east (54 min) transects.
Blue indicates free wind flow that is not influenced by wind turbines, and red indicates waked wind flow.
The density of Breckinridge and King Plains wind turbines is computed within 10 km from the PUMAS
transects. (c, f) Mean wind speed and direction profiles at each transect for parts of (blue) free and (red)
waked flows.

The mean profiles of free-flow and waked winds are shown in Figure 13c,f. Within the

turbine layer mean waked speeds were slightly (<1 m s$^{-1}$) larger than the mean free-wind values for

the EW transect (Figure 13c), contrary to expectation, but comprehensible in light of the variable

nature of the convective BL. Within 200-400 m, these profiles are the same, deviating again at

higher levels. During the WE transect (Figure 13f) both free and waked profiles were very similar.

Mean profiles of wind direction for waked and free winds are close for both EW and WE transects,

turning from 140° within the rotor layer to 175° at 1 km AGL. The statistically insignificant

difference between mean waked and free wind speed profiles in this example resulted from the

temporal evolution of winds over 55 min drive one way.





The rotor-layer-mean waked flow from the Breckinridge wind farm was 8.8 m s$^{-1}$ compared

to 10 m s$^{-1}$ of waked flow downwind of the King Plains wind farm (Figure 13b. The difference in
waked flow between the Breckinridge (8.7 m s$^{-1}$) and King Plains (8.3 m s$^{-1}$) wind farms is much
smaller on the way back (Figure 13d). As stated previously, these differences are primarily due to
the temporal variability of wind speed and a slope of a terrain along Lake Hellums Rd., which
descends from 400 m on the west to 280 m on the east.

The developed technique allows waked and free flows from measurements at different

distances from turbines to be estimated as illustrated in Figure 14 for the following transects:
(Figure 14a) within King Plains wind farm on Hwy 412; (Figure 14b) on the Breckenridge Rd.
located 0.9 km of the wind farm; and (Figure 14c) on Lake Hellums Rd. located 5 km north of the
turbines (Figure 4). Profiles show free-stream winds at locations within the wind farm 1–1.5 m s$^{-1}$
stronger than waked winds there, as expected, and the difference decreases with distance from the
farm, until at 5 km (Figure 14c), the waked and free flow profiles are equal within the standard-
deviation error. Wind directions of waked and free flows at each transect (Figure 14a–c) remain
southeast below 500 m AGL and turn to southwest at higher elevations.

Fixed sites A2, A1, and H form a south-north line through the King's Plains wind farm. The

lower panels of Figure 14 show wind profiles at these three sites averaged for three time periods
from late morning to late afternoon. For the first two time periods, the mean wind speeds at
downwind Site H were larger compared to other sites (Figure 14d, e), again contrary to expectation.
Radünz et al. (2025) also noticed this effect and attributed the differences to terrain influences that
can lead to increased wind speeds downwind.



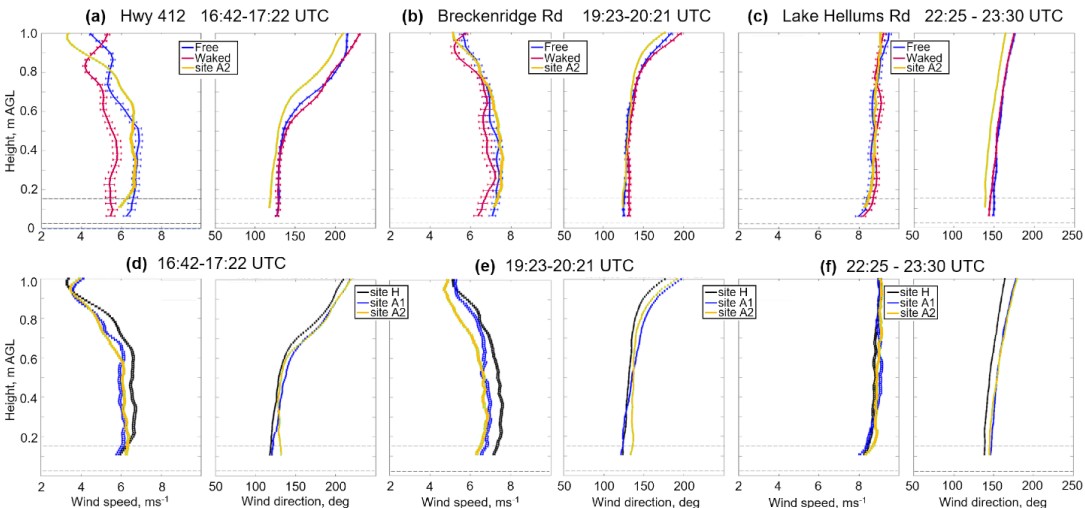

Figure14. (Top row) Mean profiles of (blue) free and (red) waked wind speed and direction from PUMAS measurements on (a) Hwy 412, (b) Breckenridge Rd., and (c) Phillips Ave. Yellow color indicates inflow wind profiles from stationary Doppler lidar at Site A2 averaged for the corresponding time interval. (Bottom row) Mean wind speed and direction profiles (d–f) from stationary Doppler lidar measurements at sites (black) H, (blue) A1, and (yellow) A2.

The technique allows us to estimate the overall impact of individual wind farms as illustrated in Figure 15. During the ~55 km transect on Lake Hellums Rd., PUMAS passed Breckinridge and King Plains wind farms twice, going east to west and back (Figure 13b, d). The difference in turbine-layer wind flow downstream of both wind farms was about 1.3 m s$^{-1}$ during the EW transect due to a slight increase of wind speeds at 22:07–22:17 UTC (Figure 15c). During the WE transect, winds downstream of both wind farms were almost equal with the mean difference of 0.36 m s$^{-1}$ (Figure 15d). Wind directions in the rotor layer were close for both transects, with differences of 2°.





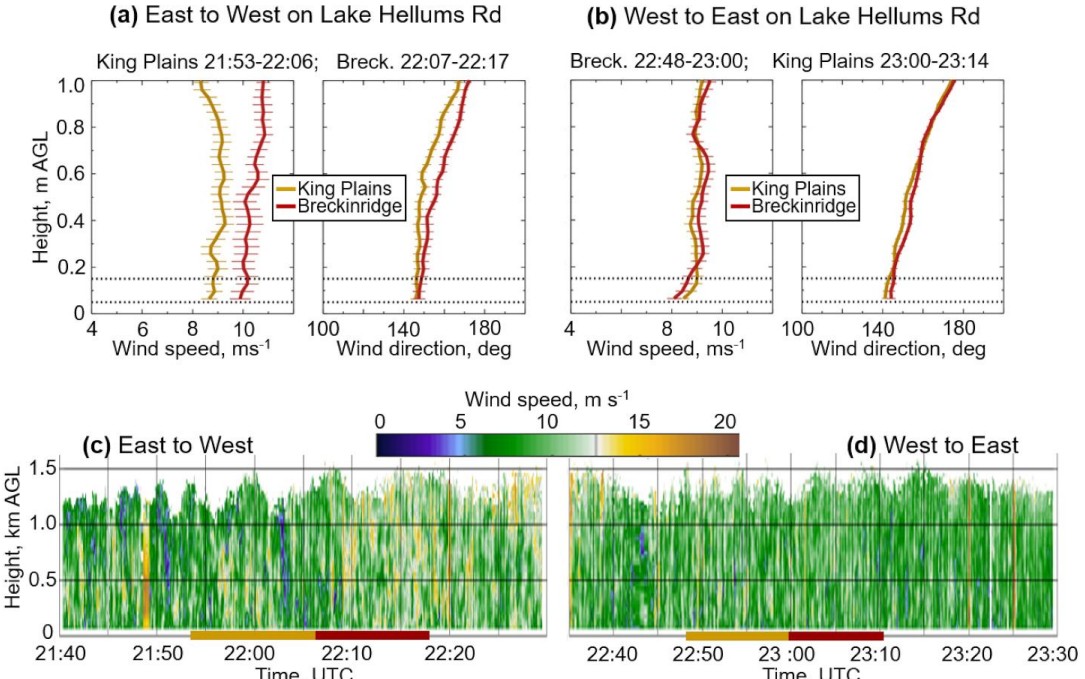

Figure 15. Wind speed and direction profiles from PUMAS measurement within the 10 km radius of influence by turbines from (gold) King Plains and (dark red) Breckinridge wind farms during (a) east to west and (b) west to east transects along Lake Hellums Rd. (Figure 13). (c, d) Time-height cross sections of wind speed at these transects. Color bars at the bottom of both panels indicate parts of each transect downstream of (gold) King Plaines and (dark red) Breckenridge wind farms.

The results in Figures 13–15 illustrate the ability to determine free and waked flows on long (>55 km) transects at various distances (0.9–5 km) from the wind farm and to compare the waked flow downwind of the Breckinridge and King Plains wind farms. These results are obtained for moderate (6–12 m s$^{-1}$) southeasterly winds and unstable BL conditions of large atmospheric variability and strong vertical mixing, leading to rapid mixing out of the wakes. Spatial variations of the free-stream wind speed, often related to small differences in terrain, and temporal changes were ~1 m s$^{-1}$, which were similar to the differences between waked and free-stream speeds, when observed. Thus, under these daytime conditions, it was often difficult to distinguish the wakes from the ambient flow. The following section will show some examples from PUMAS measurements on 5 Sep characterized by stronger (10–20 m s$^{-1}$) wind speeds.

**5.3 5 September case study, nocturnal LLJ, southwesterly winds**



Time series of wind speed from PUMAS and six stationary Doppler lidars (Figure 3) taken
at the heights closest to the turbine hub height of 90 m are shown in Figure 16 for the diurnal period
(Figure 16a) and the period of PUMAS operations (Figure 16b). Wind speed and direction from all
lidars show small differences and similar trends from sunset to midnight (0100–0600 UTC). Later
in the morning and daytime during PUMAS operations, winds at all sites fluctuate around 10–14 m
s$^{-1}$, later decreasing to 5–8 m s$^{-1}$ by the evening hours. Wind directions from all lidars show steady
turning from southeasterly (~150°) to northerly (~360°). Wind speed and direction for the period
of PUMAS measurements at 1200–1700 UTC (yellow box in Figure 16a, b) show similar variations
of data from all lidars and close period-mean data (Table 4). Slightly lower (11.6 m s$^{-1}$) mean wind
speed is observed at Site H, located in the wake of turbines for south-southwesterly directions
compared to Site D (12.9 m s$^{-1}$) and Site A2 (12.4 m s$^{-1}$) of inflow lidar measurements (Table 4).
The period-mean wind speed of 2 Hz (Figure 16b, gray) and 10-min averaged (Figure 16b, red)
PUMAS measurements are similar (Table 4) but the standard deviation of 2 HZ data is larger (2.3
m s$^{-1}$) compared to the 0.4 m s$^{-1}$ standard deviation of 10-min averaged data.

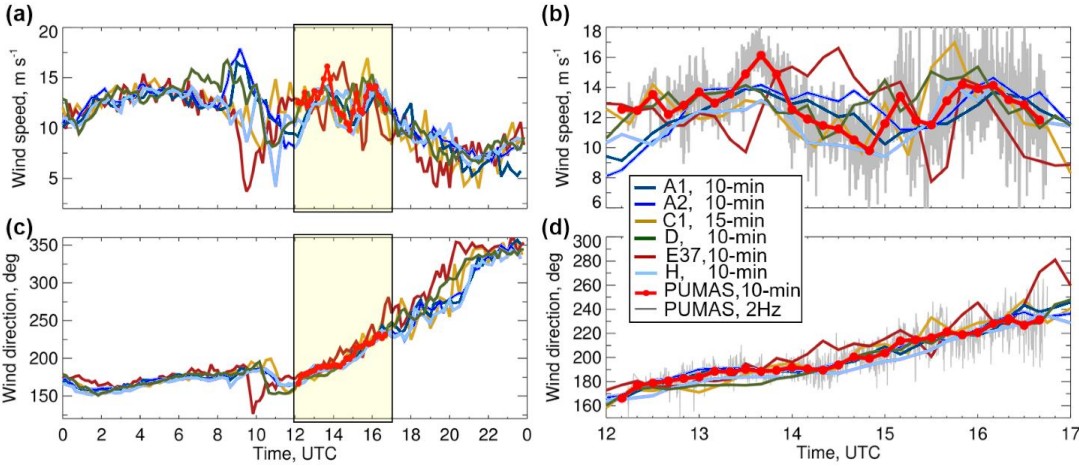

Figure 16. Wind speed and wind direction at 90–110 m from several AWAKEN stationary Doppler lidars
and PUMAS on 5 Se, 2023: (a) Time series of 10 min (15 min at C1) data from stationary lidars at several
sites are shown for 24 hours by colors according to the color scale. Data is taken close to the hub height; 90
m at sites C1, E37, D, Pumas, and 110 m at sites A1, A2, H. Yellow boxes indicate the time of PUMAS
measurements on this day. (b) Same as (a) but for the period of PUMAS measurements at 1200–1700 UTC.
The gray color indicates 2 Hz PUMAS, and the red line with dots shows 10 min averages.




Table 4. Mean and standard deviation of wind speed and direction from PUMAS and stationary

lidars over period of PUMAS operations on 5 Sep at 1200–1700 UTC.

| Site | Height | Time resolution | Speed, m s⁻¹ | | Direction, deg | |
|------|--------|-----------------|------|------|------|------|
| | | | *mean* | *STD* | *mean* | *STD* |
| PUMAS | 90 | 2Hz | 12.8 | 2.3 | 201.7 | 19.7 |
| PUMAS | 90 | 10 min | 12.8 | 1.4 | 200.2 | 18.7 |
| C1 | 90 | 15 min | 12.2 | 2.0 | 201.7 | 26.4 |
| E37 | 90 | 10 min | 12.3 | 2.4 | 209.6 | 29.3 |
| A1 | 110 | 10 min | 12.0 | 1.3 | 201.3 | 23.6 |
| A2 | 110 | 10 min | 12.4 | 1.8 | 201.7 | 21.5 |
| H | 110 | 10 min | 11.6 | 1.4 | 195.3 | 22.1 |
| D | 90 | 10 min | 12.9 | 1.2 | 198.8 | 24.5 |

An example of wind speed and direction profiles from PUMAS measurements within the
King Plains wind farm is shown in Figure 17 for three (out of 22 total) transects on 5 Sep. Transects
are shown for alternate WE and EW driving directions on Breckenridge Rd., Phillips Ave., and
Carrier Rd. (Figure 17, white arrows). The south-north distance between these roads is 1.6 km. The
length of these transects depends on road conditions and varies from 19.9 km on Breckenridge Rd.
to 12.5 km on Carrier Rd., which ends due to the terrain after crossing County Road 20.

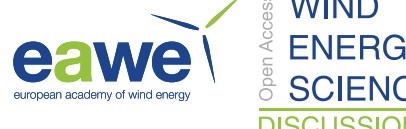

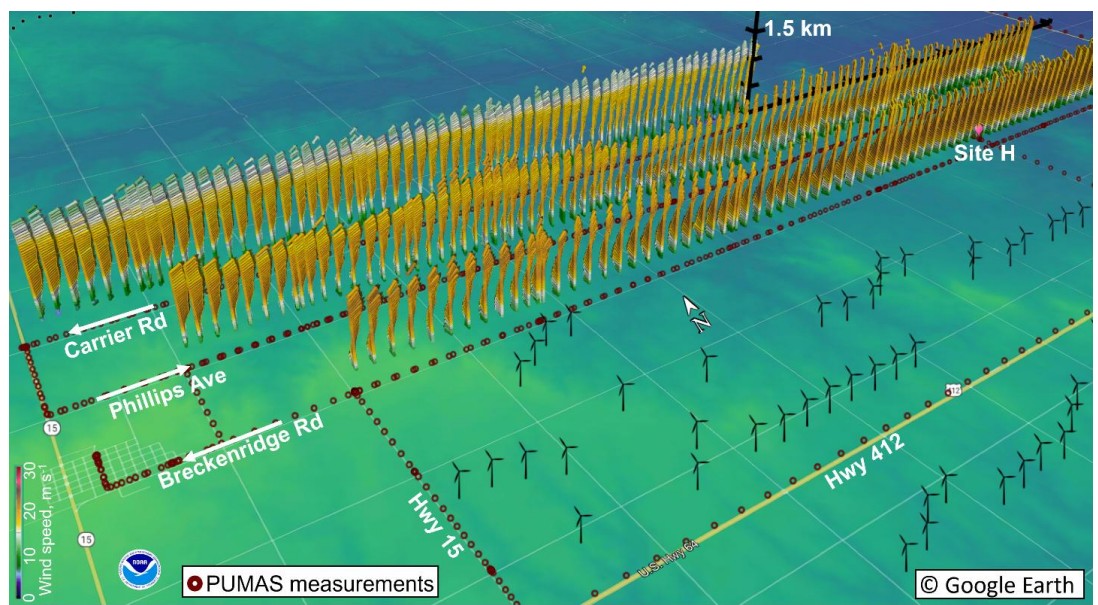

Figure 17. Wind speed (colors) and direction (arrows) profiles on 5 Sep are shown along individual transects on Breckenridge Rd. (1314–1334 UTC), Phillips Ave. (1352–1414 UTC), and Carrier Rd. (1437–1451 UTC) selected for the analysis. The dark red circles indicate points of PUMAS measurements on 5 Sep. Profiles are embedded on a Google Earth terrain elevation map (Debnath et al, 2022) and rotated clockwise ~60° for a better view. Wind speed is scaled from 0 to 30 m s$^{-1}$ according to the color scale on the left side of this figure. The horizontal distance between profiles is about 300 m. White arrows on the left corner indicate the PUMAS driving direction for each transect in this example.

Time-height cross sections (Figure 18) of simultaneously measured wind speed, wind direction, and motion-corrected vertical velocity along the waked part of the transects from Figure 17 illustrate temporal evolution of wind flows on each transect, as the convective BL mixed upward into the remaining nighttime LLJ. Wind speeds of 8–12 m s$^{-1}$ below 400 m increased to >25 m s$^{-1}$ above this height at all transects, with a strong (>28 m s$^{-1}$) LLJ within 400–600 m captured during the 20 min transect at Breckenridge Rd. The LLJ of ~25 m s$^{-1}$, observed during the 20-min transect on Phillips Ave, decreased to 20 m s$^{-1}$ at the 15 min transect on Carrier Rd. Wind directions during all transects are mostly south-southwesterly (~200°) with some episodes of southerly winds below 200 m (Figure 18b). The motion-corrected vertical velocity is weaker at Breckenridge Rd. with more downward motions (Figure 18c), but during all transects more variability is observed in the growing convective layer at low levels.




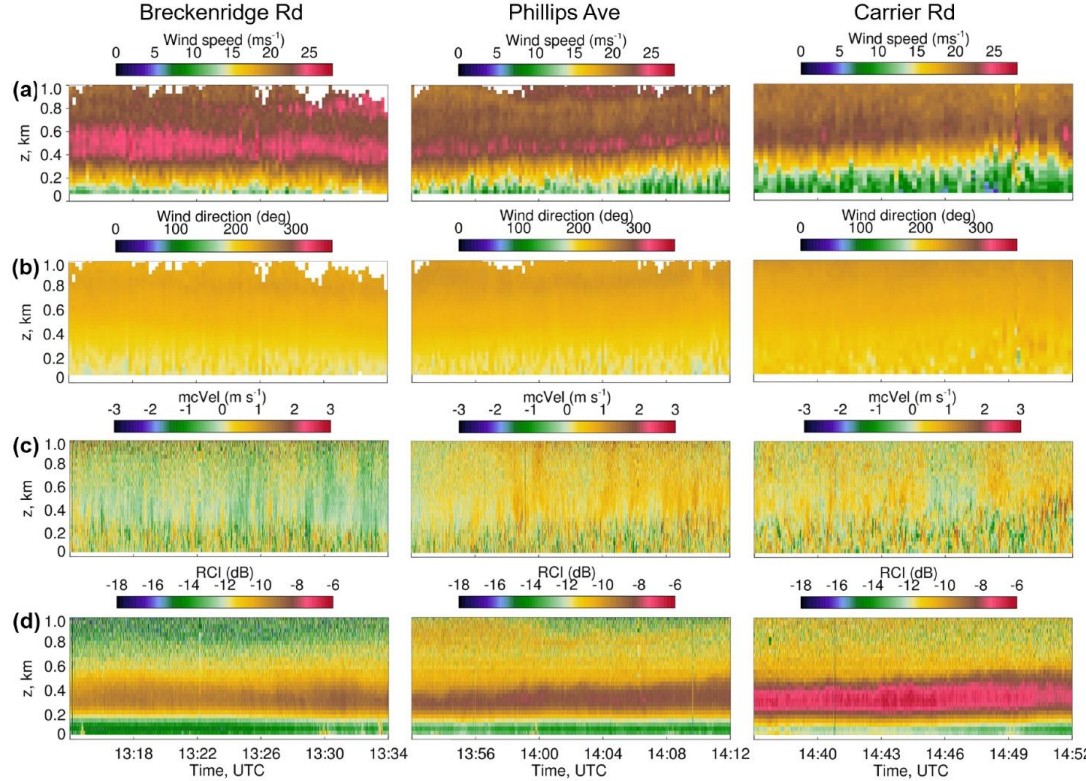

Figure. 18. Time-height cross sections of simultaneously measured (a) wind speed, (b) wind direction, (c)
motion-corrected vertical velocity, and (d) range-corrected backscatter intensity from transects shown in
Figure 17 along (left column) Breckenridge Rd., (middle) Phillips Ave., and (right) Carrier Rd. Panels c, d
are shown up to 2 km AGL to illustrate BL growth.

The temporal increase of BL depth can be seen in plots of vertical velocity (Figure 18c) and

the range-corrected intensity (Figure 18d). Measurements from stationary lidars have been used
extensively to estimate planetary boundary layer mixing height (Bonin et al. 2017), but a similar
technique using mobile lidar measurements is currently under development.

The difference between waked and free flows in the rotor layer during all transects is less

than 2 m s$^{-1}$ (Figure 19a–c), and a similar difference for the same time intervals (Figure 19d–f) is
found between wind speed measured by stationary lidars at Site A2 (inflow) and Site H (waked).
Although the mean wind direction within the rotor layer is south-southwesterly from PUMAS and
stationary lidars during all transects, the BL stability changed from stable during the transect on
Breckenridge Rd. to unstable during the transect on Carrier Rd. Wind speeds from PUMAS and



three stationary lidars decreased with time but for all periods show high shear below LLJ maxima
at 400–500 m.

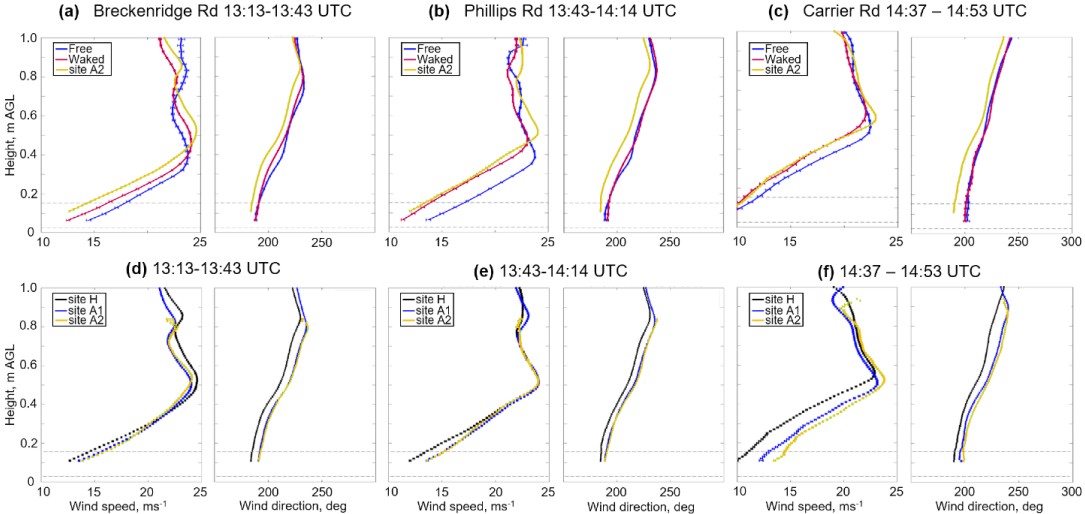

Figure. 19. Similar to Figure 14, but for the mean profiles during transects on 5 Sep shown in Figure 17. The
embedded plots indicate the BL stability.

## 6. Conclusions

Quantitative characteristics of wind and turbulence in the atmospheric layers occupied by
the wind turbine rotor blades are crucial to wind energy, as is wind information above this layer to
provide a meteorological context up to several hundreds of meters AGL. Understanding the
variability of winds across wind farms and under different conditions is a key factor in the planning
and operations of wind projects.
The high-frequency, motion-compensated PUMAS measurements of the horizontal wind
speed, wind direction, range-corrected intensity, and simultaneous vertical-velocity statistics,
including variance, skewness, and kurtosis, from a moving platform, provide a new approach to
characterizing dynamic processes critical for wind farm wake analysis. The unique PUMAS
measurements offer insight into the temporal and vertical variability of wind flows similar to
stationary scanning lidars and also reveal spatial variability of characteristics of the horizontal and
vertical structure of wind flows modified by operating wind turbines.



In the daytime convective cases studied here, spatial variations of the unwaked, free-stream
wind speeds were often ~1 m s$^{-1}$, and temporal changes along transects repeated over periods of an
hour were of similar magnitude. Differences in waked vs. free-stream speeds, when discernable,
were also ~1 m s$^{-1}$, so it is often difficult to distinguish turbine or wind-farm wake effects from the
natural atmospheric variability under these conditions.
Data from the mobile lidar can also complement the AWAKEN instrumentation to
understand the effect of a large wind farm on wind flows under different background wind
conditions and stratification. The PUMAS measurements can be used to evaluate wind simulation
by models and improve wake model prediction accuracy. The truck-based mobile Doppler lidar
data analyses show that advances in measuring, understanding, and modeling the atmospheric
boundary layer within wind farms will be required to provide improved meteorological support for
wind energy.
The developed technique allowed the sampling and automated analysis of wind speeds
influenced by wind turbine clusters located at different distances from PUMAS transects and the
flexibility to adjust the sampling drive patterns to account for any wind directions.
**Author contribution**: YP and AB planned the PUMAS measurement campaign, processed and
analyzed the data; BMC, MH, and RM operated the mobile lidar and performed the
measurements; MZ provided remote software support, YP wrote the manuscript draft; RB, ES,
SB, and BC reviewed and edited the manuscript, SL and NB provided data from stationary lidars
and edited the manuscript; PM planned the overall Awaken campaign.
**Acknowledgment.** The authors thank the AWAKEN experiment participants who aided in the
deployment and the collection of remote sensing data and our colleagues who monitored, quality
controlled, and provided data to the Data Archive. Funding was provided by the U.S. Department
of Energy Office of Energy Efficiency & Renewable Energy Wind Energy Technologies Office.,
The mobile Doppler lidar measurements in Oklahoma, as part of the AWAKEN field was
supported by the National Oceanic and Atmospheric Administration (NOAA) Atmospheric
Science for Renewable Energy (ASRE) program and by. This research the NOAA cooperative
agreement NA22OAR4320151, for the Cooperative Institute for Earth System Research and Data
Science (CIESRDS). This work was authored in part by the National Renewable Energy Laboratory
for the U.S. Department of Energy (DOE) under Contract No. DE-AC36-08GO28308. Funding was



provided by the US Department of Energy Office of Energy Efficiency and Renewable Energy
Wind Energy Technologies Office. The views expressed in the article do not necessarily represent
the views of the CIESRDS, NOAA, DOE or the U.S. Government. The U.S. Government retains
and the publisher, by accepting the article for publication, acknowledges that the U.S. Government
retains a nonexclusive, paid-up, irrevocable, worldwide license to publish or reproduce the
published form of this work, or allow others to do so, for U.S. Government purposes. We thank
Amy Brice from NREL for editing the paper according to the journal requirements.
**Data availability statement**. All the data are publicly available. Datasets from scanning Doppler
lidars at the Atmospheric Radiation Measurement (ARM) Southern Great Plains (SGP) sites C1
and E37 are available from the ARM SGP Archive at
https://www.arm.gov/capabilities/observatories/sgp. Data from scanning Doppler lidars operated
during AWAKEN experiment are available from the Atmosphere to Electrons Wind Data Hub
(https://www.a2e.energy.gov, U.S. Department of Energy, 2024). The lidar data DOI at site A1 is
https://doi.org/10.21947/2375440, at site A2 is https://doi.org/10.5439/1890922 , at site H is
https://doi.org/10.21947/2283040, and at site D is https://doi.org/10.21947/2375440.
The sonic anemometer data DOI at site A2 is https://doi.org/10.21947/2375440. And the NREL
ASSIST thermodynamic profiler at site B is https://doi.org/10.21947/2375440.

**Appendix A**. CSL/NOAA Field Projects in 2018–2024 using mobile lidar systems.
Table A1. Mobile Doppler lidar measurements from various platforms.

| Platform | Project | Date | Location |
|---|---|---|---|
| Aircraft & Truck | Utah Summer Ozone Study (USOS) | July-Aug 2024 | Salt Lake City, Utah |
| Aircraft | Airborne Methane Mass Balance Emissions in Colorado (AMMBEC) | July 2024 | Front Range, Colorado |
| Aircraft | Airborne and Remote sensing Methane and Air Pollutant Surveys (AiRMAPS) | 2024 | U.S. East Coast |
| Truck | Oil and Gas Air Quality Study (DJ-CDPHE II) | Oct-Nov 2023 | NW Colorado |
| Truck | American Wake Experiment (AWAKEN) | Aug-Sep 2023 | Central Oklahoma |





| Aircraft | Coastal Urban Plume Dynamics Study (CUPiDS ) | June-Aug 2023 | New York City Region |
|---|---|---|---|
| Truck | Pilot Studies in Colorado Front Range (PUMAS) | Feb-Mar 2023 | Metro Denver, Colorado |
| Aircraft & Truck | California Fire Dynamics Experiment (CalFiDE) | Aug-Sep 2022 | California & Oregon |
| Aircraft | System Integration and Test Experiment (SITE) | Jun-Aug 2021 | Florida |
| Aircraft & Truck | Southwest Urban NOx and VOC Experiment (SUNVEx) | Aug 2021 | Las Vegas, Nevada, Louisiana, & California |
| Truck | Oil and Gas Air Quality Study (DJ-CDPHE I) | Sep 2021 | Metro Denver, Colorado |
| Truck | Pilot Studies in Colorado Front Range (PUMAS) | Oct-Nov 2021 | Metro Denver, Colorado |
| Truck | Pilot Studies in Colorado Front Range (PUMAS) | Jun-Oct 2020 | Metro Denver, Colorado |
| Ship | Atlantic Tradewind Ocean-Atm. Interaction Campaign (ATOMIC) | Jan-Feb 2020 | Tropical North Atlantic |
| Aircraft | Fire Influence on Regional to Global Env.& Air Quality (FIREX-AQ) | Jul-Aug 2019 | Pacific Northwest |
| Aircraft | Fire Winds (FIREWinds) | Jun 2018 | Florida |
| Ship | Propagation of Intra-seasonal Tropic Oscillations (PISTON) | Aug-Oct 2018 | Philippine Sea |






**Appendix B. Test-drives around wind farms in Colorado**


Several test-drives of PUMAS were performed around wind farms in Sterling and Limon
located in the northern and southern parts of Colorado (Figure B1 a, b) to obtain information on
system performance, measurement errors, and driving strategies. The data were used to establish
measurement capability to study dynamic processes upwind and downwind of turbines. Figure B1
c shows motion-stabilized vertical velocity obtained from a lidar beam pointing zenith (90$^\circ$
elevation angle).

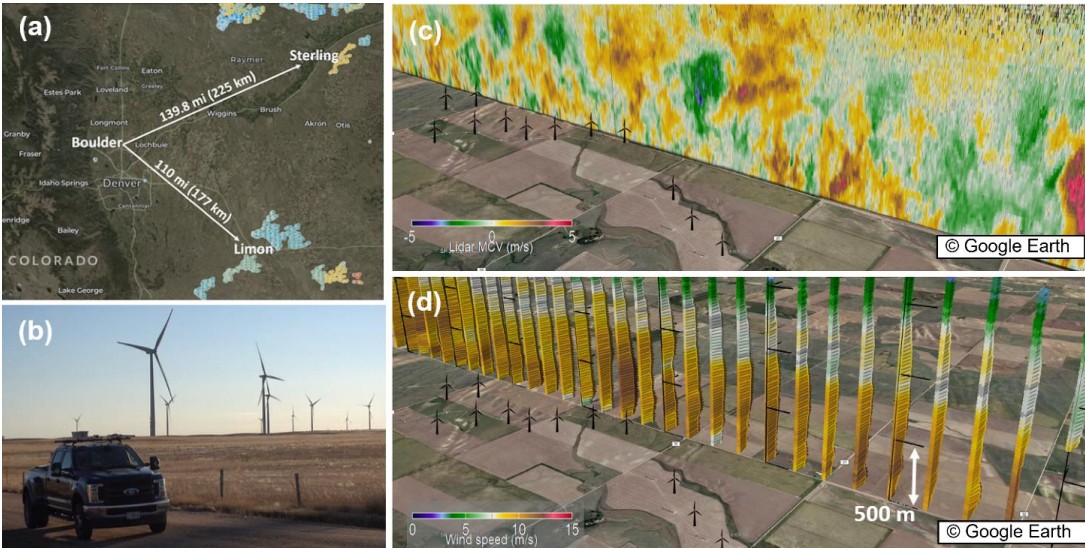


Figure B1. (a) A USGS map of wind farms located ~200 km to the northeast (near Sterling) or to the
southeast (near Limon) from Boulder, selected for PUMAS test drives in 2020, 2021, and 2023 (Table A1,
Appendix A); (b) a picture of PUMAS driving in the vicinity of wind turbines; (c) Profiles of vertical velocity
along a (~22 km) path are shown on Google Earth; (d) Profiles of wind speed (colors) and wind direction
(arrows)  along the same path. Black horizontal lines indicate height increments of 500 m.
The high temporal (~20 s) and vertical (30 m) resolution of these profiles yields unique
information about the extent and strengths of the vertical motions, including thermal updrafts and
turbulence at the cloud base. Measurements from conical scanning at 15° from the zenith (Figure
B1 d) reveal southerly wind speeds of ~12 m s$^{-1}$ up to 1.5 km.





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
