# Peer review of "Emerging mobile lidar technology to study boundary-layer"

_Wind Energy Science, 2025_

## Author Comment (AC2)

Comments on the manuscript entitled "Emerging mobile lidar technology to study boundary-layer winds influenced by operating turbines" submitted to WES

The authors reported a mobile lidar technology (PUMAS) for characterizing wind flows, which has the potential to become a powerful tool for wind assessment in the design and operation of wind energy projects. Measurement results were presented in the paper, demonstrating the capability of the proposed technology. It is an interesting and important work.

The following concerns require proper attention before the manuscript can be accepted for publication.

1. The atmospheric flow evolves as the measurement is taken on a moving truck. There is a characteristic timescale for a specific atmospheric flow event, while the measurement introduced another timescale determined by the speed of the moving truck and the flow event of interest. Choosing a proper path and driving speed seems to be important to ensure the effectiveness of the measurement. What are the authors' thoughts on this issue? What approaches are being taken to cover enough spatiotemporal range in the proposed measurement system? Are there any best practices for measuring typical flow phenomena, e.g., LLJ, wind turbine wakes, and atmospheric flows in complex terrain?

1). The reviewer points out an important interpretation issue, the parsing of along-track variations into spatial and temporal components. We have discussed this in previous papers (Pichugina et a. 2012; Banta et al. 2013b) but did not go into enough detail here, in part because our daytime operations and the relative constancy of the well-mixed boundary layer flow made these issues less of a problem for these analyses. They are, however, important considerations for mobile-platform sampling in general, so we have added a paragraph to the Introduction addressing this point. Specifically for this study, when operating the mobile lidar we used 10–20-minute repeat legs as a guideline to minimize confusion from spatial and temporal evolution. I.e., driving on the King Plains roads eastward and then back westward to the same point took less than 20 minutes so that the atmosphere did not have much time to change temporally, and our measurements were thus more representative of spatial changes rather than temporal. Of course, this assumes some stationarity of the atmosphere, but this is a reasonable assumption for most of the AWAKEN cases given the meteorology and terrain of the area. The longer transects, usually 50-60 min were from the round trips between the hotel, and the AWAKEN study area also consist of two parts: influenced by wind turbines and a free flow (see Fig. 3). That being said, the overarching temporal evolution across hours spent measuring in the wind farms on a given day was much more likely to be subject to large-scale changes that may mask the smaller-scale processes that we are interested in. This is where the other AWAKEN measurements, such as wind profiles upwind of all wind farms from stationary Doppler lidars as well as from the long-term lidar measurements from the nearby ARM SGP sites C1 (15-min data) and E37 (10-min data), are useful for context. Besides, all available NWP models were monitored before the final planning of PUMAS measurement during each day. In some cases, if any, larger-scale temporal changes may be removed as a time-dependent mean flow, and turbine wakes may be represented as anomalies.

Regarding the "best practices for measuring typical flow phenomena, e.g., LLJ, wind turbine wakes, and atmospheric flows in complex terrain?". The new text in the Introduction has recommended repeating sampling tracks and mixing in fixed-sensor data into the analyses and

interpretation of the data, and these should be incorporated as key aspects of best practices for the use of mobile sensors for reasons described. The PUMAS operated during AWAKEN over a short period of mid-August-mid-September due to the involvement of the instrument into other NOAA projects. Besides measurements were taken mostly on the late morning-daytime hours to obtain the communication and support (if needed) from the engineers in the office located in a different time zone. The measurement conditions were characterized by low (3-4 m/s) to moderate (10-12 m/s) winds and a rare late-morning remnant of the nocturnal LLJ was observed on Aug 5 and analyzed in the paper. As mentioned in the paper, our goal was to test the instrument along the performance of a motion compensating system in the real conditions driving within wind farm, develop a better driving pattern, and prepare for the future wind experiments.

2. Interpretation of PUMAS measurements is not straightforward. The measurements contain distributions of physical quantities in both space and time directions. The authors plot most of their results in the time-height cross sections. I understand that this is partly for comparison with the measurements from the stationary lidar. On the other hand, one advantage of PUMAS is that it provides variations of wind in space (in both vertical and horizontal directions). Can the obtained measurements be employed to show the spatial variation (not in the vertical direction) of a flow phenomenon, say, the downwind variation of wind turbine wakes (the authors showed some results, but the capability does not seem to be well demonstrated)?

2). Yes, as pointed in the paper, these time-height cross sections represent 3d wind variability, vertical, temporal and spatial. In this paper we did not emphasize the spatial variability along each transect due to the short length of the transects, flat terrain, and more generally due to daytime turbulence masking the signature of turbine wakes. We have done this successfully for other projects in the past, and as in the replies to the previous comment, we have added a paragraph to the Introduction to address these issues.
The added paragraph is "*Profile measurements from a moving platform document the horizontal variability of the flow, which could (for example) be due to turbine wakes or terrain-related flows, within a curtain of data along the track. But also included is variability due to temporal changes during the transect (Pichugina et al. 2012). For instance, a frontal passage halfway through a sampling leg will appear as a difference between the first and second half of the leg. Lacking additional information, one cannot determine whether these measurements show a genuine, persistent difference in the flow between the two regions. Other small-scale phenomena over the sampling track at timescales smaller than the sampling time interval of the leg may similarly appear to be horizontal variations. One approach for clarification is to retrace the path, as in the offshore LLJ example of Pichugina et al. (2012: see their Fig. 15 and accompanying text), to look for persistence of flow structures, indicating stationarity. Another is to use a mix of mobile and fixed-platform sensors to sort out the spatial and temporal variabilities, as proposed by Banta et al. (2013). In the following we use both approaches*"

3. On lines 657-660, the authors stated the capability of the proposed PUMAS technology in predicting flow statistics of different orders. However, the paper mostly focused on the first-order statistics. It is necessary to examine the proposed PUMAS system in measuring higher-order statistics, like variance, skewness, and energy spectrum.

3). The PUMAS lidar measurements can be processed to examine higher orders, similar to a stationary Doppler lidar. We have not examined the obtained higher orders of turbulence such as variance and skewness in this article as we expect or already know that turbine wake effect would just be masked by daytime turbulence. The rich dataset obtained can be used for more analysis and future research papers

4. Some discussions are necessary on the uncertainty of the measurements, like how the measurement accuracy depends on the atmospheric conditions, terrains, and the measuring conditions of the PUMAS itself.

4). The general information on the uncertainty of the measurements was provided in the paper. As mentioned, not much variations in wind conditions and terrain were observed during PUMAS measurements. But all data including as time-series of pitch, roll, lidar height (ASL), measured and motion corrected vertical velocity are available for a future detailed analysis.

5. In the conclusions section, it is suggested to discuss the limitations of the proposed PUMAS technology and potential issues to be addressed.

5). Ideally it would be great to obtain long-term measurements over various seasons and atmospheric events. We do not see much limitation for PUMAS measurements except the very nasty road conditions for driving the truck.

---

## Author Response (AR1)

**Review** of "Emerging mobile 1 lidar technology to study boundary-layer 2 winds influenced by operating turbines"

The paper is motivated to demonstrate that the PUMAS system can be a powerful and promising measurement device. The technical design of the system is presented with good details and clear narrative.

The selection and interpretation of examples from the measurement campaigns, however, do not yet bring enough confidence in PUMAS as a tool for quantitative wind resource assessment or wind farm effects studies. Therefore, while the technical development is promising, the scientific analysis and validation are not yet sufficient for publication in their current form. A revised manuscript focusing on measurement validation and clearer data–physics consistency would be of value.

*Reply*: The PUMAS was developed mainly for Air Quality tasks, but we had the opportunity to test the instrument for wind measurements around wind farms. We are fully satisfied with results of our objectives including testing the performance of the motion compensation system, the ability to obtain continuous measurements during various times of the day and various roads conditions. We also were able to compare data obtained in stationary position and while moving. Also, we want to bring to your attention the following:

1. The selected period of PUMAS measurements during AWAKEN dictated by the time and crew availability between two major CSL/NOAA experiments. That is why it was hard to find a significant difference in the meteorological conditions between measurement days. All days were mostly like 7 Sep with low winds and relatively steady wind directions. The 5 Sep was different, as clearly shown by stationary lidars. So, we concentrated on these days.
2. We continued the instrument updates such as developing lower-angle scans compared to the existing 15 degree of zenith and adding the ability of RHI (slice) scans.
3. Yes, the data do not "bring enough confidence in PUMAS as a tool for quantitative wind resource assessment or wind farm effects studies", but it is a first experiment, and measurements during stronger winds or over different seasons are needed. Also the results will be used to develop better driving pattern. To our knowledge, the development and updates of stationary lidars to provide fully reliable, remote data took about a decade.

**Specific comments**

The interpretation of vertical profiles in at least one case (Section 4.2) appears to be inaccurate and should be carefully revisited.

*Reply*: Thanks for pointing this out, we revisited and corrected this section as follows:
1. Lines 426-432 are removed from the text.

2. We've gotten rid of the L plots (Fig. 10 e, f) that didn't seem to make sense, and the relevant stability information is shown by virtual temperature from the ASSIST retrievals.
3. Line 437. The text "(e, f) Obukhov length from the PNNL flux station at Site A2 for these days" is removed from the Fig. 10 caption.

Moreover, the connection between Section 4.2 and Chapter 5 is unclear—Chapter 5 seems to begin with a dynamic interpretation of measurements already presented in Section 4.2.

*Reply*: Section 4 provides the context of the measurements from the (traditional) fixed-sensor point of view. To clarify it we changed name of Section 4 (Line 374) to "**4. 5 and 7 September case studies: Fixed-site context measurements**".

1. Section 4.2 discusses the TROPoe retrievals from thermodynamic profiler (ASSIST) for two days (Sep 5 and Sep 7)
2. Section 5 presents the mobile PUMAS analyses and discussion.

We have added a sentence at line 378 to clarify this: ... stability. In this section we characterize the boundary layer evolution these days based on fixed-location sensor measurements. Figure 8 shows ...

1. P4, L106: The phrase "... decreases the weight and the size of both modules" is unclear. Presumably, the two-part design connected by an umbilical cord makes each module lighter and easier to handle, although the total system mass may remain similar or even increase. Please clarify whether this is the intended meaning and how the umbilical design reduces the total weight, if at all.

*Reply:* The actual phrase in question reads, "This design, along with significant decreases in the weight and size of both…" which we feel does make it clear that the novel aspects of this lidar system are both that it is separated into two smaller modules and that each module has been made more compact, and, therefore, that the mass of each component (and thus of the total system) has been reduced. Besides, the two-part design allowed better use of space, for example inside the aircraft.
1. Here is a quote from Schroeder et al, 2020, who lead this design "The new instrument has enabled greater flexibility in field campaigns where previous instruments would have been too costly or space prohibitive to deploy".
2. Line 108. similar "design" is changed in the text to similar "capability"

2. P20-21, sec. 4.2, Fig. 10: The classification of 7 September as unstable does not appear to be supported by the data. A Monin–Obukhov length near zero is inconsistent with the strongly positive vertical gradient of potential temperature. Furthermore, the similar behavior of the ABL top on both days suggests that conditions were not drastically different. This interpretation should be reviewed. A simple plot of the potential temperature profiles (e.g., at 10 UTC for both days) would be very helpful. Also, check whether panels (e) and (f) might have been inadvertently swapped.

*Reply:* Yes, there appear to be issues with the Obukhov lengths L in Fig.10e-f—thank you for pointing this out. We have deleted these two panels from Fig.10 and eliminated the discussion of them from the text. Our discussion of these plots applies only to PUMAS measurement period (shaded in gray on Figs.10c and d), and we have added the parenthetical phrase "(gray shading)" to the text to make this more clear. We believe the rest of the text accurately describes conditions during those periods, but we have added the following sentence (Line 423) to further highlight important aspects of the data, as suggested by the reviewer. A new sentence is added after words for 5 Sep and 7 Sep. "The time-height cross sections show cooler temperatures near the surface prior to 16 UTC and warmer daytime surface temperatures after 17 UTC, and also the growth of the convective layer (black line) after 15 UTC, on both days."  Stability estimates ...

3. P25, L514-517: This statement is puzzling—it implies that the experimental setup may be inadequate to provide data consistent with physical expectations. The same issue arises in the discussion around Figure 14. The authors' explanations of these differences seem somewhat ad hoc and do not convincingly account for the observed inconsistencies.

What we are really seeing is that the significant variability in the ambient flow, due to the strong turbulence in the daytime convective boundary layer and the variations of topography along the track of measurements, is larger than the horizontal variations due to turbine waking, which are also being reduced by rapid mixing out by the turbulence. The result is that horizontal variations due to the wakes are mixed out and jumbled up with turbulence and terrain variability, so that often one can't tell the difference, or sometimes even counterintuitive effects may be seen. We have amended the text to make these points better.

**Citation**: https://doi.org/10.5194/wes-2025-79-RC2\.

---

## Author Response (AR2)

Dear Editor,

 Please find below our responses to your comments

**Public justification (visible to the public if the article is accepted and published)**:
For the most part the authors have addressed the comments from the reviewers, but there are a couple of issues that I think still need to be addressed related to the new paragraph in the Introduction, the limitations of the study and future research. I will inform the authors and reviewers.

Additional private note (visible to authors and reviewers only):
For the most part the authors have addressed the comments from the reviewers, but there are a couple of issues that I think still need to be addressed.
1. Responses to R1 mention "we have added a paragraph to the Introduction addressing this point" and "The new text in the Introduction has recommended repeating sampling tracks and mixing in fixed-sensor data into the analyses and interpretation of the data, and these should be incorporated as key aspects of best practices for the use of mobile sensors for reasons described". I can't find the added paragraph you mention "Profile measurements from a moving platform ..." Please check this as the Introduction of v4 seems to be the same as v1. Should there be a later version than v4?

The missing paragraph was added after Line 83.

"Profile measurements from a moving platform document the horizontal variability of the flow, which could (for example) be due to turbine wakes or terrain-related flows, within a curtain of data along the track. But also included is variability due to temporal changes during the transect (Pichugina et al. 2012). For instance, a frontal passage halfway through a sampling leg will appear as a difference between the first and second half of the leg. Lacking additional information, one cannot determine whether these measurements show a genuine, persistent difference in the flow between the two regions. Other small-scale phenomena over the sampling track at timescales smaller than the sampling time interval of the leg may similarly appear to be horizontal variations. One approach for clarification is to retrace the path, as in the offshore LLJ example of Pichugina et al. (2012: see their Fig. 15 and accompanying text), to look for persistence of flow structures, indicating stationarity. Another is to use a mix of mobile and fixed-platform sensors to sort out the spatial and temporal variabilities, as proposed by Banta et al. 2013 In the following we use both approaches."

2. You mention in response to reviewers,

a. "It is a first experiment, and measurements during stronger winds or over different seasons are needed.

b. "Also, the results will be used to develop better driving pattern".

c. " We have not examined the obtained higher orders of turbulence such as variance and skewness in this article as we expect or already know that turbine wake effect would just be masked by daytime turbulence. The rich dataset obtained can be used for more analysis and future research papers", d. "all data including as time-series of pitch, roll, lidar height (ASL), measured and motion corrected vertical velocity are available for a future detailed analysis."

e. "Ideally it would be great to obtain long-term measurements over various seasons and atmospheric events."

I think these are useful additions to the manuscript in terms of a section on limitations of the study and future research. Can you add a paragraph based on these comments (a -e)?

The sub-section is added to the text after line 678:

*Limitations of the study and future research.*

The results presented in this paper are obtained from the short pilot study, mostly during daytime hours and low wind conditions. Therefore, we have not examined the obtained higher orders of turbulence such as variance and skewness in this article as we expect or already know that turbine wake effect would just be masked by daytime turbulence. Anyway, the rich dataset obtained can be used for more analysis and future research papers, the development of longer transects and driving patterns around wind farms. In addition, all data including as time-series of pitch, roll, lidar height (ASL), measured and motion corrected vertical velocity are available for a future detailed analysis. Ideally it would be great to obtain long-term measurements over various seasons and atmospheric events.

3. Line 108 missing full stop after capability.

The dot after the word capability was added.

4. Line 39. "Cupp 1990_," was changed to " Cupp 1990,"

---

## Author Response (AR3)

Dear Editor,

According to your remarks, the accepted version was checked, and the final version is uploaded. Some minor changes are tracked in the uploaded Word file